# Ce-mediated molecular tailoring on gigantic polyoxometalate {Mo$_{132}$} into half-closed {Ce$_{11}$Mo$_{96}$} for high proton conduction

Xue-Xin Li[1,3], Cai-Hong Li[1,3], Ming-Jun Hou[1,3], Bo Zhu[1], Wei-Chao Chen[1]✉, Chun-Yi Sun[1], Ye Yuan[1], Wei Guan[1], Chao Qin[1], Kui-Zhan Shao[1], Xin-Long Wang[1]✉ & Zhong-Min Su[1,2]

Precise synthesis of polyoxometalates (POMs) is important for the fundamental understanding of the relationship between the structure and function of each building motif. However, it is a great challenge to realize the atomic-level tailoring of specific sites in POMs without altering the major framework. Herein, we report the case of Ce-mediated molecular tailoring on gigantic {Mo$_{132}$}, which has a closed structural motif involving a never seen {Mo$_{110}$} decamer. Such capped wheel {Mo$_{132}$} undergoes a quasi-isomerism with known {Mo$_{132}$} ball displaying different optical behaviors. Experiencing an 'Inner-On-Outer' binding process with the substituent of {Mo$_2$} reactive sites in {Mo$_{132}$}, the site-specific Ce ions drive the dissociation of {Mo$_2^*$} clipping sites and finally give rise to a predictable half-closed product {Ce$_{11}$Mo$_{96}$}. By virtue of the tailor-made open cavity, the {Ce$_{11}$Mo$_{96}$} achieves high proton conduction, nearly two orders of magnitude than that of {Mo$_{132}$}. This work offers a significant step toward the controllable assembly of POM clusters through a Ce-mediated molecular tailoring process for desirable properties.

Polyoxometalates (POMs), a unique subset of molecular clusters, possess diverse structural types and potential applications from biomedicine to catalysis and materials science[1–8]. Recently, targeted design and synthesis of POM clusters have attracted much interest to explore the basic principles of structure and bonding as well as to expand the functionalities of POM-based materials and devices[9–12]. Advances have been conducted in terms of diverse nuclearities, sizes, and shapes; for instance, the general building block strategy with 'bottom-up' molecule design[13] has enabled the assemblies of POM architectures, especially giant POM entities that bridge the gaps between molecular oxo anions and infinite oxides[14]. However, the controllable preparation of giant POM clusters, including precise atomic-level tailoring and structural functionality, still remains a great challenge, which is highly desirable for engineering functional

molecular metal oxides and a fundamental understanding of how different motifs contribute to their overall properties[14,15]. The bottleneck mainly relies on the limited type of giant POM clusters that can be used as rational tailoring precursor: (1) it should have available reactive or clipping building block sites that are favorable for the site-specific tailoring to target POM assemblies without changing the whole structure; (2) wheel- or cage-shaped giant POMs may be ideal models as they could provide confined and sizable environments for the molecular container (e.g., water vapor adsorption) and proton transport purposes[14,16,17]. Therefore, the exploration of eligible giant parent POM clusters and tailor-made derivatives for unusual functionality is a striking task.

Polyoxomolybdate 'big wheels' are a well-researched iso-POM family[17] constructed from fundamental repeating {Mo$_{11}$} building

[1]Key Laboratory of Polyoxometalate and Reticular Material Chemistry of Ministry of Education, Department of Chemistry, Northeast Normal University, Ren Min Street, No. 5268, Changchun, Jilin 130024, P.R. China. [2]State Key Laboratory of Supramolecular Structure and Materials, Institute of Theoretical Chemistry, College of Chemistry, Jilin University, Changchun, Jilin 130021, P.R. China. [3]These authors contributed equally: Xue-Xin Li, Cai-Hong Li, Ming-Jun Hou. ✉e-mail: chenwc061@nenu.edu.cn; wangxl824@nenu.edu.cn

blocks, namely mix-valenced $\{Mo_8\}$-linked by $\{Mo_2\}$- and $\{Mo_1\}$-type units, as presented in the archetypal tetradecameric $\{Mo_{154}\}$[18] and hexadecameric $\{Mo_{176}\}$[19]. In principle, other oligomeric assemblies (e.g., decamer, 10 sets of $\{Mo_{11}\}$ units) along with the inherent open cavity of Mo wheels were possible to be formed by grouping a number of common building blocks. Such Mo wheels could be further modified by Mo-based capped fragments to afford cage-like architectures, the merely reported example is $\{Mo_{248}\}$ that undergoes a molecular growth process with $\{Mo_{176}\}$ as the parent scaffold[19]. Of particular importance are the bimetallic $\{Mo_2\}$ units in Mo wheels; it is relatively reactive and can be easily eliminated, coordinated by carboxylic acids, or replaced by electrophiles, to produce Mo wheel species with 'defect' sites[20], amino acid-[21] or lanthanide (Ln)-functionalization[22], respectively. At this level, it is reasonable to speculate that the doping of Ln centers into the Mo wheel is favorable to regulate the ring shapes and sizes if both the coordination modes and positions of Ln centers on the wheels could be manipulated in a controlled manner. Therefore, we considered the cage-like Mo wheels involving enough $\{Mo_2\}$ sites to be suitable giant POM precursor model for exploring the possible Ln-mediated precise atomic-level tailoring process. What's more, giant POM isomers or quasi-isomers, which remain little reported, are ideal models for the investigation of structure–property relationships in metal oxide materials[23]. Inspired by this, we report $\{Mo_{132}\}$ (1) with a closed structural motif arising from the symmetrical growth of two $\{Mo_{11}\}$ caps located on both sides of a highly expected but never seen $\{Mo_{11}\}_{10} = \{Mo_{110}\}$ decamer. The discovery of capped wheel 1, along with the classical $\{Mo_{132}\}$ ball[24], constitutes a unique pair of structural quasi-isomers in POMs that present different optical behaviors. Especially, a Ce-mediated molecular tailoring process was demonstrated through the substituent of edge-sharing $\{Mo_2\}$ reactive sites by Ce centers featuring overall 'Inner-On-Outer' binding modes, which drives the resection of one $\{Mo_{11}\}$ cap of 1 through the site-specific dissociation of $\{Mo_2^*\}$ clipping sites and finally gives rise to a predictable half-closed $\{Ce_{11}Mo_{96}\}$ (2) product (Fig. 1). The enhanced proton conductivity between both clusters arising from Ce-mediated tailoring process has been fully examined and discussed.

## Results

The decameric 'pure' molybdenum framework 1 was prepared by reducing an acidified aqueous mixture of $Na_2MoO_4·2H_2O$, $Na_2S_2O_4$, and $CH_3COONa·3H_2O$ under hydrothermal condition, while the tailoring product 2 was obtained through the reaction of an aqueous solution of 1 with cerium chloride (Related discussions had been added into the synthesis discussion part in Supplementary information, Supplementary Fig. 1 and Supplementary Table 1). Both compounds were characterized crystallographically, and their formula assignments (as below) are fully verified through an array of techniques as confirmed in the giant wheel- and ball-shaped polyoxomolybdates[17], such as single crystal X-ray structure analysis, elemental analyses, redox titration, UV/Vis spectroscopy as well as bond valence sum (BVS)

calculations (see Supplementary information for detailed discussion, Supplementary Tables 2–7 and Supplementary Figs. 2–4):

$Na_{22}H_{30}\{Mo_{132}O_{372}(OH)_{10}(H_2O)_{12}(SO_4)_5(CH_3COO)_{20}\}·122H_2O \equiv Na_{22}H_{30}\{1a\}·122H_2O$

$Na_3H_6\{Ce_{11}Mo_{96}O_{286}(H_2O)_{101}(SO_4)_8\}·170H_2O \equiv Na_3H_6\{2a\}·170H_2O$

## Structural analysis of Mo₁₃₂

The single-crystal X-ray structural analysis (Supplementary Table 8) displays that 1 crystallizes in the tetragonal space group $P4_2/ncm$. It not only possesses a classical wheel topology with a $D_{5d}$ symmetry but also has two caps grafted to both rims of the wheel, thus affording a closed $\{Mo_{132}\}$ capped wheel (Fig. 2). In this sense, polyoxoanion 1a could be divided into two parts, one is the main $\{Mo_{110}\}$ wheel (Fig. 2a) composed of $\{Mo_8\}$, $\{Mo_2\}$, and $\{Mo_1\}$ building blocks, each occurring ten times, and the others remain two $\{Mo_{11}\}$ caps (Fig. 2c). Remarkably, $\{Mo_{110}\}$ wheel contains the fundamental $\{Mo_{11}\}$ building blocks in Mo wheel family (Supplementary Fig. 5) including theoretical '$\{Mo_{132}\}$' model, archetypal $\{Mo_{154}\}$ and $\{Mo_{176}\}$ discovered by Müller et al.[24], which constructs a pure decameric Mo wheel reported to date. It should be noted that $\{Mo_2\}$ units between two neighboring $\{Mo_6\}$ pentagons are rather different in the known Mo wheels: the edge-sharing $\{Mo^V_2\}$ units in 1a have a shorter Mo···Mo distance (ca. 2.55 Å) compared to corner-sharing $\{Mo_2\}$ units in other oligomers (Fig. 2g). Each Mo atom of the edge-sharing $\{Mo_2\}$ makes use of both oxygen atoms from equatorial plane and axial positions for binding to relative outer rim defined by adjacent pentagons of the wheel framework (Fig. 2g). The edge-sharing $\{Mo_2\}$ units are the common building blocks for Mo ball[24] and highly reduced Mo POMs[25,26]. Such major structural difference originated from $\{Mo_2\}$ units, giving rise to a tight $\{Mo_{110}\}$ Mo wheel (ring diameter: ca. 26 Å, Supplementary Fig. 6a) than expected. For this reason, this decameric pure molybdenum framework reported herein is the smallest available Mo wheel of this type, as proposed by Cronin et al.[27]. As a matter of principle, the hypothetical dodecameric '$\{Mo_{11}\}_{12}$' member with a mixture of both edge-sharing and corner-sharing $\{Mo_2\}$ units may also be found experimentally in the near future (Supplementary Fig. 5). The Mo wheel family may serve as an ideal model to understand the phenomena in confinement situations in terms of their selectable internal nanospaces.

On both sides of the $\{Mo_{110}\}$ wheel are capped by two pentagonal $\{(Mo)Mo_5\}$-containing $\{Mo_{11}\}$ units (Fig. 2h). It is interesting to observe the in-plane growth of two $\{Mo_{11}\}$ caps occurs along the direction of inner rim defined by $\{Mo^V_1\}$ groups (denoted as $\{Mo_1^*\}$, Fig. 2e and Supplementary Fig. 7, belong to the $\{Mo_8\}$ units in $\{Mo_{110}\}$ wheel) between the adjacent $\{Mo_2\}$ units in $\{Mo_{110}\}$ wheel. Both capping moieties are accordingly turned at an angle of 36° relative to one another (Fig. 2b), in line with the corresponding relative positions of $\{Mo_1^*\}$ units. Such $\{Mo_1\}$-driven in-plane growth process in 1a is quite different from the corner-sharing $\{Mo_2\}$-mediated in-plane growth ($\{Mo_{248}\}$)[19] or longitudinal growth process ($\{Mo_{180}\}$)[20] (Supplementary Fig. 8). In this respect, Mo wheels are favored to trap specific molybdenum fragments/intermediates (e.g., $\{Mo_{17}\}$[21], $\{Mo_{30}\}$[20], and $\{Mo_{36}\}$[19]) that can't be survived in separate form in ring rims or centers for closed cage-like or host-guest architectures, respectively. The main interception driving force, such as hydrogen bonding or coordination bonds, comes from the simple Mo-based species themselves and their organic derivatives on the inner rims of the wheels. In 1a, this covalently bonded Mo-based cap unit with the composition $\{(Mo^VO)_5(Mo^{VI})Mo^{VI}_5O_{21}(H_2O)_6\}$ is not unique since it has been used for essential groups in building Keplerate $\{Mo_{132}\}$ ball[24]. Thus, 1a represents the fundamental building blocks of both the Keplerate Mo sphere and Mo wheels simultaneously. Furthermore, another kind of edge-sharing $\{Mo^V_2\}$ units (denoted as $\{Mo_2^*\}$, Fig. 2e and Supplementary Fig. 7) have been realized during the combination of the wheel and corresponding caps. One Mo atom within the cap and above mentioned $\{Mo_1^*\}$ in wheel contribute to this $\{Mo_2^*\}$ unit with its four O atoms in the

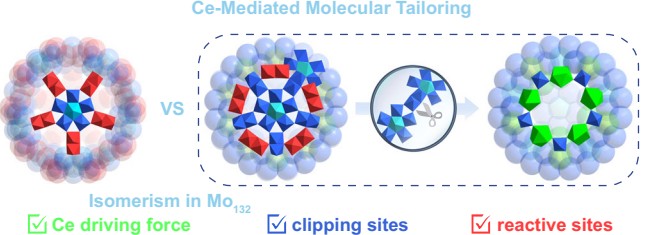

**Fig. 1 | Ce-mediated molecular tailoring on gigantic $\{Mo_{132}\}$ polyoxomolybdate.** Ce-mediated molecular tailoring from a gigantic $\{Mo_{132}\}$ capped wheel POM featuring structural quasi-isomerism with known $\{Mo_{132}\}$ ball to a predictable half-closed $\{Ce_{11}Mo_{96}\}$ product. Color code: Mo blue/red/cyan/yellow, Ce green.

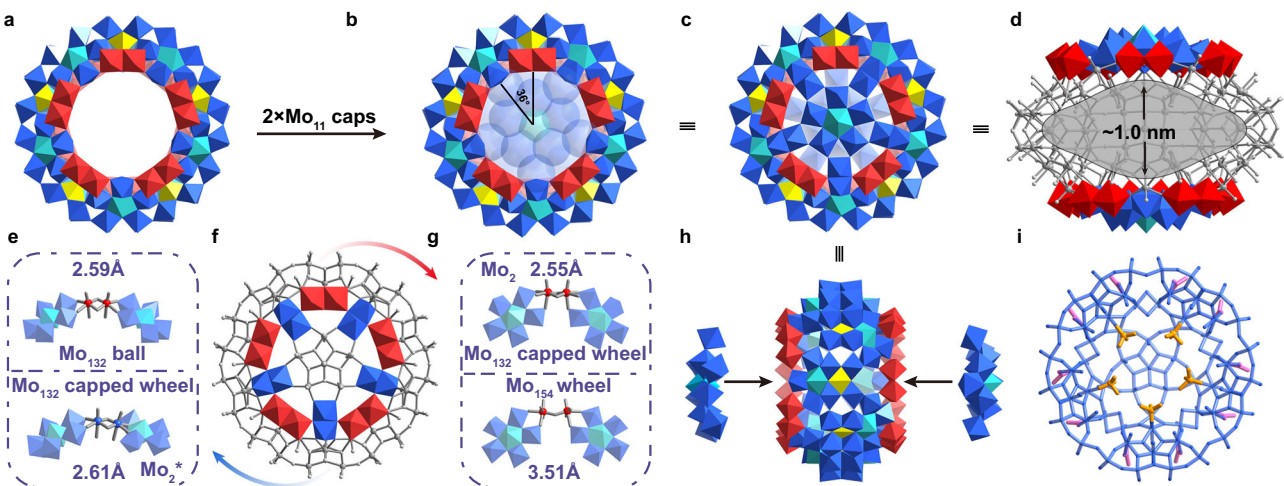

**Fig. 2 | The structure of gigantic {Mo₁₃₂} capped wheel.** Polyhedral and ball-and-stick representations of **a** {Mo₁₁₀} wheel, **b** the angle of capping moieties, **c 1a**, **d** the cavity (shown gray) in **1a**, **e**–**g** two types of edge-sharing {Mo₂} units, **h** side view of **1a** and **i** anisotropic ligand distributions in **1a**. Color code: Mo blue/red/cyan/yellow, S orange, O gray, C pink.

equatorial planes for connectivity (Fig. 2e). Therefore, the slightly bent {Mo₂*} facilitates an asymmetrical curved growth along the equatorial plane (Fig. 2e) to afford 'oblate spheroid' topological structure of **1a** that achieves a cavity with the maximum internal diameter *ca.* 1.0 nm (Fig. 2d). We find that the assembly modes of edge-sharing {Mo₂*} units reported here point the way toward a fresh type of combination between two {Mo₆} pentagons compared to that in spherical Keplerate {Mo₁₃₂} cluster[24], whose {Mo₂} mainly boosts the symmetrical growth as a function of linkers. It follows that two types of edge-sharing {Mo₂} units with unequal connectivity during the assembly lay the foundation of the formation of **1a** (Fig. 2f).

In general, small ligands (e.g., sulfates, phosphate, or carboxylate-containing species) are prone to replace coordinated water molecules on the inner surface of {Mo₂} in Keplerate {Mo₁₃₂} capsule[28] and Mo wheels[29] for effective stabilization or functionalization. This behavior is also presented in cage-like **1a** but with anisotropic ligand distributions (Fig. 2i). The sulfate internal ligands, each with half site-occupancy disorder (5 {SO₄²⁻} in total) are found to be attached to 10 well-constructed {Mo₂*} units (Supplementary Fig. 9a) while acetate external ligands (20 {CH₃COO⁻} in total) are grafted onto both sides of {Mo₁₁₀} wheel with the 20 positions consisting of two Mo adjacent atoms came from {Mo₂} units and {Mo₆} pentagons (Supplementary Fig. 9b). The current ligand self-sorting of a rigid cage-like entity may give **1a** to mimic biological behavior such as small molecular recognition or responsive sensing. After the successful substitution of ligated water, the bidentate carboxylate here not only exerts a forceful constraint on the Mo atoms for pushing them to be much closer to each other but also offers the possibility of external groups modification in polyoxomolybdates, which is expected to consider **1a** as a network synthon for preparing POMs-based high-dimensional supramolecular frameworks[30].

## Isomerism in 132-nuclearity POMs: Mo Brown vs Mo Green

Exploring the molecular isomerism phenomenon is a significant task that allows for understanding possible assembly mechanisms and overall properties[31,32], but such molecular isomerism in POMs is still rare[23,33]. Herein, the development of the Mo-based building blocks assembly strategy brings the isomerism in 132-Mo-atom clusters in the case of their identical reduction degree (45%, Moⱽ₆₀Moⱽᴵ₇₂) and characteristic {Mo(Mo)₅} motifs with similar connection rules. As shown in Supplementary Fig. 10, **1a** contains 12 {Mo₆} pentagons as well as 10 edge-shared {Moⱽ₂} linked by other simple Mo species that give rise to a capped wheel **1a** while the spherical {Mo₁₃₂} is built up of 12 {Mo₆}

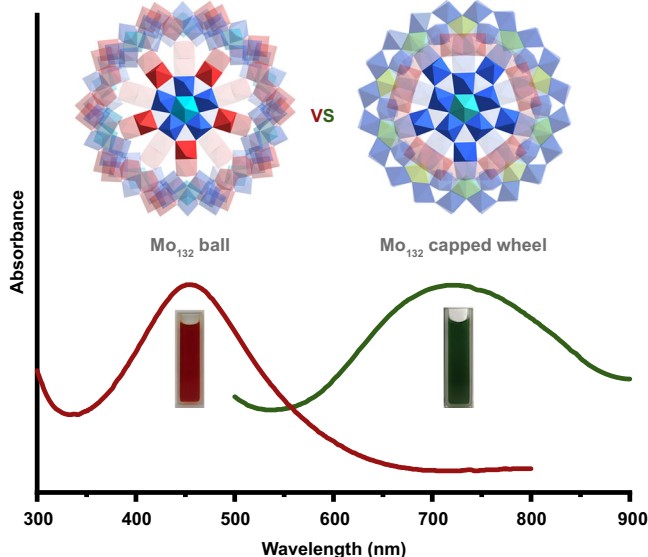

**Fig. 3 | Characteristic UV/Vis spectra in 132-nuclearity POMs.** Capped wheel {Mo₁₃₂} and Keplerate-type {Mo₁₃₂} ball. Color code: Mo blue/red/cyan/yellow. Source data are provided as a Source Data file.

pentagons and 30 edge-shared {Moⱽ₂} units, hence, they represent quasi-isomers (the descriptions about the definition of such quasi-isomers as well as the comparison of their synthesis conditions had been presented in Supplementary Fig. 10 and related discussions). This molecular quasi-isomerism offers a model for knowing how different motifs within the 132-Mo-atom POMs contribute to their overall properties, for example, UV/vis absorption spectra in solution and the solid state.

Figure 3 shows **1a** displays an intense dark green color with an absorption peak in the long wavelength range (*ca.* 720 nm), which remains different from the classical {Mo₁₅₄} wheel featuring blue color with the characteristic absorption peak located *ca.* 750 nm. In addition, the Keplerate-type {Mo₁₃₂} reveals a brown color and possesses an absorption peak in the short wavelength range (*ca.* 450 nm) in the UV/vis spectra in solution[25]. The reason for different UV/Vis absorption in solution is the delocalization of the reducing electrons. It is well-known that {Mo₁₅₄} wheel (Mo Blue) contains the electrons from reduction delocalized over the structure in the central belt spanning

the ring. As for the {Mo$_{132}$} ball (Mo Brown), the electrons are localized in reduced and isolated {Mo$^V_2$} units, which is further confirmed by the electron density map and molecular orbitals of {Mo$^V_2$}-containing {Mo$_{16}$} unit in {Mo$_{132}$} ball using density functional theory (DFT) calculations (Supplementary Fig. 11 and related discussions). When the reducing electrons are delocalized not only over the ring but also the isolated {Mo$^V_2$} units, this contributes to the capped wheel **1a** defined as a 'Mo Green' family. The electron density maps and molecular orbitals of reasonable calculation models in **1a**, including {Mo$_5$} species (Supplementary Fig. 12 and related discussions) in wheel and {Mo$_{26}$} species (Supplementary Fig. 13 and related discussions) that contains reduced {Mo$^V_2$} units, highlight that the distributions of reducing electrons are localized in {Mo$_5$O$_6$} double cubane of ring and isolated {Mo$^V_2$} units as expected, respectively. Therefore, from this pair of 132-Mo-atom clusters, we found that the color displayed by reduced giant polyoxomolybdates could not only be related to the degree of Mo reduction but also the factor of the location of reduction.

Moreover, the solid UV/vis absorption spectra of both {Mo$_{132}$} clusters also show different absorption bands (Supplementary Fig. 14a). **1** exhibits a much wider absorption range, including UV and visible. On this basis, the diffuse reflectance spectrum shows that the corresponding well-defined optical band-gap energies of **1** can be assessed at about 1.13 eV (Supplementary Fig. 14b). Meanwhile, the relevant conduction band of **1** was also tested by Mott–Schottky measurements at different frequencies, which features a smaller $E_g$ (Supplementary Fig. 14c) and higher conduction band (Supplementary Figs. 15, 16) compared to {Mo$_{132}$} ball. In total, this pair of 'molybdenum framework isomers' exhibits disparate optical behaviors, which may act as cluster models that bridge the gap between molecular isomerism and phase isomerism of molybdenum oxide for the purpose of better understanding both structures and photochemical performance of phase-dependent nanomaterials[34].

## Ce-mediated tailoring product {Ce$_{11}$Mo$_{96}$}

The stability of giant capped wheel **1** in solution is necessary to be confirmed as the precision during the tailoring process can't be separated from the whole integrity of this parent. Both UV/Vis and Raman spectra before and after tetrabutylammonium (TBA) salt exchange of **1** is conducted for this purpose. The dark green precipitation of TBA-**1** was obtained by the mixture of a diluted aqueous solution containing **1** as well as an aqueous solution of TBABr (Supplementary Fig. 17). UV/vis absorption spectrum of **1** in aqueous solution remains almost coincident for 24 h (Supplementary Fig. 18). Moreover, the unanimous absorptions and profiles of UV/vis absorption spectra of **1** in water and TBA-**1** in acetonitrile solution imply the good stability of **1** (Supplementary Fig. 18). As for the Raman spectrum of **1**, two characteristic peaks at 992 and 818 cm$^{-1}$ could be attributed to the vibrations of terminal Mo=O bands and Mo–O–Mo bridges (Supplementary Fig. 19), respectively[26,35]. In addition, all scattering peaks had been maintained in the Raman spectrum of TBA-**1**, which shows that the structure of **1** was kept during the cation exchange process. The stability of **1** encourages us to employ this cage-like giant POM cluster featuring abundant edge-shared {Mo$_2$} units as a suitable precursor for exploring the rational molecular tailoring in the presence of cerium centers. It is possible to drive the tailoring process at the building-block level when Ce species were used as strong electrophiles to replace {Mo$_2$} units on the parent {Mo$_{110}$} wheel. Fortunately, we were able to get the half-closed Ce-containing product **2** after a careful optimization of the reaction conditions. Single-crystal X-ray structure analysis shows that **2a** crystallizes in *Pnma* space group and displays poor aqueous solubility. All {Mo$_2$} units on the {Mo$_{110}$} wheel had been replaced by cerium ions that give rise to such a half-closed motif with one {Mo$_{11}$} cap missing (Fig. 4a, b). Meanwhile, the substitution of eleven Ce centers greatly reduces the symmetry of **2a** ($C_{2v}$ point group) as compared to tailoring parent **1a** ($D_{5d}$ point group). The half-closed

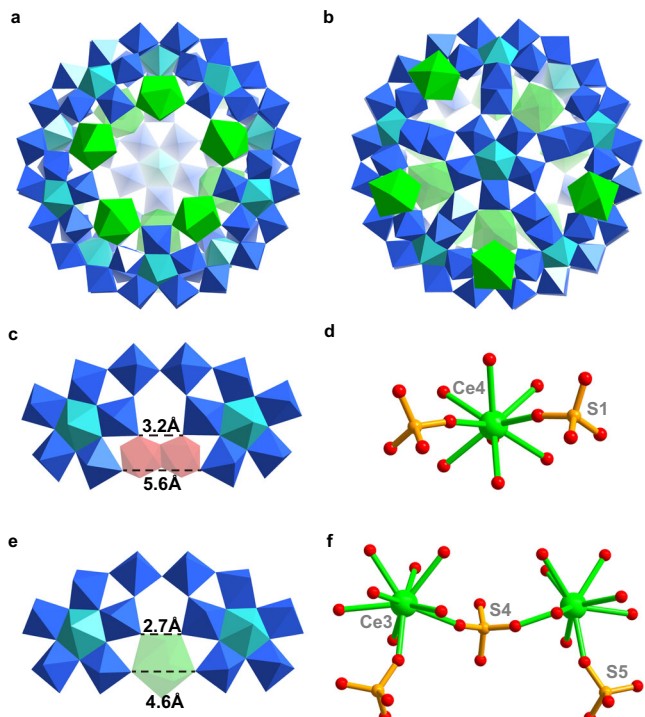

**Fig. 4 | The structure of Ce-Mediated Tailoring Product {Ce$_{11}$Mo$_{96}$}.** Polyhedral representations of **a** front-view and **b** back-view of **2a**; Comparison of the distances between equivalent O atoms for two adjacent {Mo$_8$} units in **c 1a** and **e 2a**; **d** Ball-and-stick representations of [{CeO$_5$(H$_2$O)$_4$}(SO$_4$)$_2$] and **f** [{CeO$_4$(H$_2$O)$_4$}$_2$(SO$_4$)$_3$] guest fragments inside **2a**. Color code: Mo blue/red/cyan, Ce green, S orange, O red.

motif based on Ln–Mo POMs is rather rare since the only one remains, the {Mo$_{130}$Ce$_6$}, suffers from a molecular growth of the hypothetical '{Mo$_{120}$Ce$_6$}'[20]. By contrast, the half-closed **2a** here is derived from a well-defined **1a** with a predictable Ce-mediated molecule tailoring process.

From the crystal structure, as shown in Fig. 4, this Ce-mediated molecule tailoring arises from the Ce centers adopting the 'Inner-On-Outer' binding modes related to different coordination environments (Fig. 5). (1) Ce 'On' binding modes (Fig. 5a). In total, there are five Ce centers (Supplementary Table 9, Ce–O bands, 2.474(10)–2.673(14) Å) located just on the position of {Mo$_2$} reactive sites of the wheel. Each features the same 4-connected rod as presented in {Mo$_2$} units linked with neighboring {Mo$_6$} pentagons. From the respective distances between relevant {Mo$_8$} fragments corners across the {Mo$_2$} or Ce bonding sites for the tailorable **1a** and half-closed **2a**, we can find that the overall distances between equivalent O atoms pairs are reduced slightly (from 5.6 to 4.6 Å for the inner rim and from 3.2 to 2.7 Å for the belt polyhedra) since the monatomic {Ce(H$_2$O)$_3$} and {Ce(H$_2$O)$_5$} moieties are a litter smaller than the bimetallic {Mo$_2$} unit (Fig. 4c, e). It is, therefore, reasonable to prove that the five 4$f$ centers all adopting Ce 'On' binding modes cause a certain shrinkage of the rim of the wheel and hence the missing of the {Mo$_{11}$} cap on one side. This key point has been well confirmed by the neutral Ln-substituted Mo ring with no caps as a result of the same Ln 'On' binding manner[27]. (2) Ce 'Inner' (Fig. 5b), 'Outer' (Fig. 5d) as well as 'Inner–Outer' (Fig. 5c) binding modes (Supplementary Table 9, Ce–O bands, 2.469(13)–2.68(2) Å). The other 4$f$ substitution sites are responsible for why the {Mo$_{11}$} cap on the other side is kept retained. {Ce(H$_2$O)$_4$} moiety with Ce 'Inner' binding mode and {Ce(H$_2$O)$_7$} moieties with Ce 'Outer' binding modes attach to the inner and outer surfaces of the half-closed oblate spheroid so as to replace {Mo$_2$} units, respectively. Furthermore, Ce 'Inner–Outer' binding modes are defined as a set of two Ce centers that are located at both the inner and outer surfaces of the major

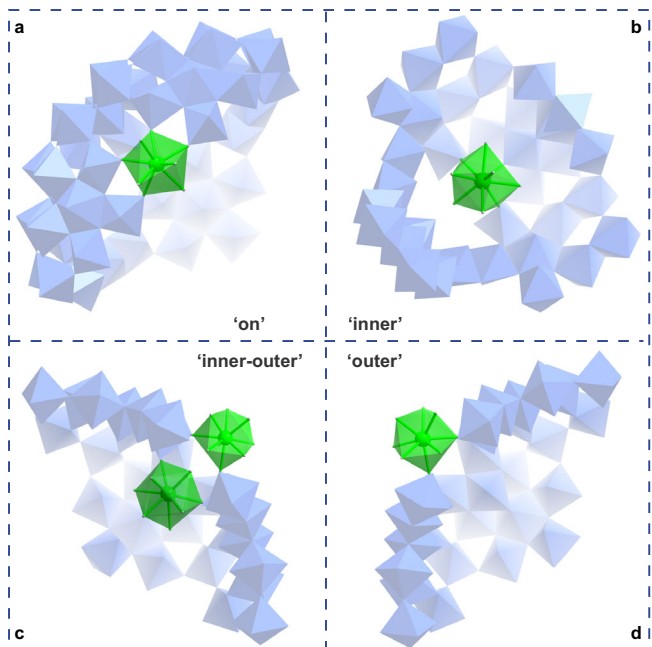

**Fig. 5 | The 'Inner-On-Outer' binding modes of Ce centers in 2a.** Polyhedral representations of **a** Ce 'On' binding modes, **b** Ce 'Inner' binding modes, **c** Ce 'Inner–Outer' binding modes, and **d** Ce 'Outer' binding modes. The Ce and Mo centers are shown in the green and blue polyhedra, respectively.

molybdenum framework and take the place of {Mo₂} units. All outer {Ce(H₂O)₇} moiety act as 2-connected rods while all inner {Ce(H₂O)₄} moiety act as 3-connected rods (Supplementary Fig. 20). To the best of our knowledge, Ce 'Inner–Outer' binding modes had not been reported in 4f-doped gigantic Mo wheels[20,22]. Specifically, the aforementioned distances between equivalent O atoms pairs have not changed, leading to the retainability of {Mo₁₁} cap in **2a**. In terms of the small ligands in **2a**, 3 {$\mu_2$-$\eta^1$:$\eta^1$:$\eta^0$:$\eta^0$-SO$_4^{2-}$} groups are decorated onto Ce centers with Ce 'On' binding modes for further stabilization of 4f ions, the remaining {SO$_4^{2-}$} groups (Supplementary Table 10, S-O bands, 1.438(11)–1.613(18) Å) still adhere to {Mo$_2^*$} units as presented in **1a** while all grafted {CH₃COO} groups disappear on account of Ce substitution. In fact, the inside sulfates bring great influence during the assembly of Ce centers featuring Ce 'Inner' binding modes. One Ce ion seems to be 'trapped' into half-closed **2a** by the linkage of two identical {SO$_4^{2-}$} groups (Fig. 4d) through two Ce–O–S bridges (Ce4–O244, 2.458(10) Å; S1–O244, 1.461(11) Å), and the other 'trapped' Ce ions disordered over two positions (half site-occupancy disorder in crystallography) possess the alternate linkage (Fig. 4f) guided by three {SO$_4^{2-}$} groups (the central one adopts a $\mu_4$-$\eta^1$:$\eta^1$:$\eta^1$:$\eta^1$ bridging mode) through four Ce–O–S bridges (Ce3–O99, 2.450(15) Å; Ce3–O181, 2.546(14) Å; Ce3–O99, 2.450(15) Å; S4–O181, 1.448(13) Å; S5–O99, 1.520(15) Å). Based on this, both linkages contribute to the successful encapsulation of [{CeO₅(H₂O)₄}(SO₄)₂] and [{CeO₄(H₂O)₄}₂(SO₄)₃] guest fragments inside half-closed **2a** (Fig. 4d, f).

All eleven Ce$^{III}$ ions are nine-coordinated with the +3 oxidation state testified by X-ray photoelectron spectroscopy (XPS) analysis (see support information for all Mo$^{V/VI}$, S$^{VI}$, and Ce$^{III}$ XPS in **1** and **2**, Supplementary Figs. 21–25) that could be described as distorted monocapped square antiprism[22]. It can be seen that the coordination modes and positions of Ce ions had been effectively controlled, and the expected tailoring product, namely half-closed **2a**, was obtained. In this regard, it appears that **1a** remains a rather suitable parent in directing the formation of tailoring product **2a** owing to its two types of edge-sharing {Mo₂} units. The {Mo₂} units of {Mo₁₁₀} wheel are prone to be replaced by Ce centers with the Ce 'On' binding modes, which

provide the driving force for the occurrence of molecule tailoring process, while the {Mo$_2^*$} units consisting of two Mo atoms from the wheel and cap could be separated play the part of clipping sites to access the site-specific resection of {Mo₁₁} cap (Fig. 1 and Fig. 2f). The molecule tailoring on **1a** not only brings the losing of {Mo₁₁} cap but also prompts the missing of five Mo atoms of the wheel without altering the main molybdenum framework (Supplementary Fig. 26). Also, the BVS calculations[27,36] of Mo-based double cubes and UV/vis spectrum (the characteristic band is centered around 730 nm) suggesting 20 reducing electrons are delocalized throughout the Mo wheel structure as expected.

**Proton conductivity**

Recently, POMs have emerged as advanced solid proton conductors in light of their structural diversity with oxygen-rich surface and chemical stability, etc[37–39]. **1** and **2** possess abundant H₂O molecules, rich H⁺ counter cations, terminal oxygen atoms, and high stability towards hydrous conditions; at this level, we choose proton conductivity tests to evaluate the functionality of **1** and **2** related to Ce-mediated precise tailoring strategy. Both compacted pellets of crystalline powder samples were used for alternating current (AC) impedance measurements. Various parameters, including relative humidity (RH, 60–98%) and temperature (30–80 °C), had been fully explored and the resulting performance implies the molecular tailoring product **2** displays a rather better proton conductivity than parent **1** (Fig. 6). At 30 °C and 60% RH, the conductivity value of **2** was measured to be $4.02 \times 10^{-6}$ S cm$^{-1}$ (Fig. 6a). The conductivity has rapidly grown to $2.21 \times 10^{-2}$ S cm$^{-1}$ upon increasing RH to 98% (Fig. 6b). Similar situation also occurs in **1** (Fig. 6b and Supplementary Fig. 27) with conductivity ranging from $5.90 \times 10^{-7}$ S cm$^{-1}$ (60% RH) to $2.41 \times 10^{-4}$ S cm$^{-1}$ (98% RH). In general, such highly humidity dependent proton conductivity has often been discovered in POMs-based solid proton conductors as the water molecules play a major role during proton movement process[40,41]. This is in line with the water vapor adsorption of **1** and **2** that increase with the rising humidity, as demonstrated by the room-temperature H₂O vapor absorption and desorption isotherms (Fig. 6c). The difference in humidity-dependent proton conduction in both compounds may be identified as half-closed motif **2** possesses a better water uptake capacity (312.52 cm³ g⁻¹) compared to closed motif **1** (160.59 cm³ g⁻¹). As the temperature increase from 30 to 80 °C, the proton conductivity increases with the increasing temperature. This may be attributed to the possible promotion of forming hydronium ions based on H₂O and H⁺ at elevated temperatures and thus accelerate the proton transition[42]. The conductivity of **1** under 98% RH gradually raises and reaches a maximum of $1.17 \times 10^{-3}$ S cm$^{-1}$ (Fig. 6e and Supplementary Fig. 28). Interestingly, the bulk conductivity of **2** increases very slowly from $2.21 \times 10^{-2}$ S cm$^{-1}$ to $9.01 \times 10^{-2}$ S cm$^{-1}$ with the temperature elevated from 30 to 80 °C at 98% RH (Fig. 6d, e), suggesting less temperature dependent proton conductivity, which is a rare example of metal-oxo clusters-based proton conductors that keep high and stable proton conductivity over a relatively wide temperature region[43,44]. The maximum proton conductivity of **2** close to the order of 10⁻¹ S cm⁻¹ features one of the representative POMs-based crystalline proton conductors[45–48] (Supplementary Table 11) and is comparable to those of well-known crystalline metal–organic (e.g., MOF-808-4SA-150[49], BUT-8-(Cr)A[50], MOF-808C[51], and covalent–organic frameworks (e.g., H₃PO₄@NKCOF-1[52], iCOFMs-SO₃H[53]). The activation energy ($E_a$) at 98% RH for the proton transfers in **1** and **2** is calculated to be 0.29 eV and 0.38 eV, respectively (Fig. 6e), based on the Arrhenius equation, indicating the Grotthuss mechanism (typically $E_a < 0.4$ eV) by non-diffusive routes, that is a proton hops alone between a proton donor and an acceptor without a molten carrier[54]. Fast proton conduction in the best-known Nafions under low-temperature and high RH could also be recognized as this mechanism[55]. The electron conductivity (direct-current experiment) and ionic conductivity

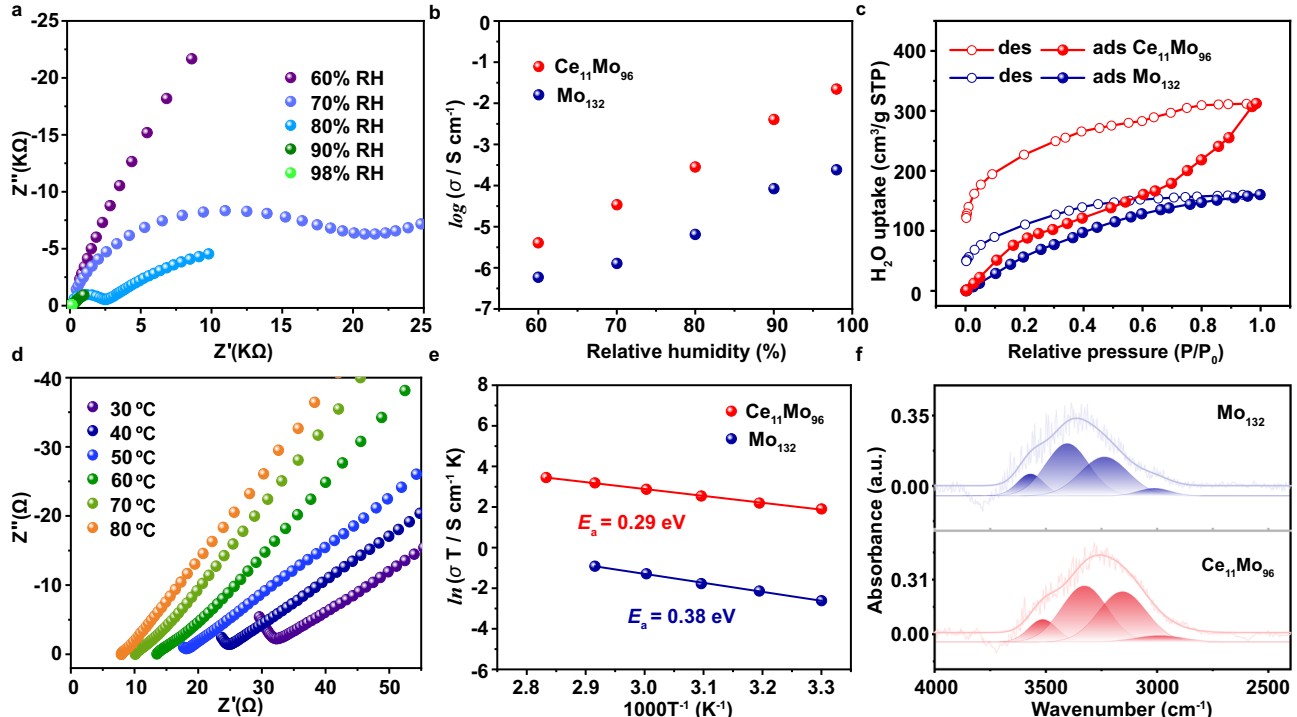

**Fig. 6 | Proton conductivity at various RH and temperatures. a** The Nyquist plots of **2** at 30 °C and various RH (60–98%). **b** Humidity-dependent proton conductivity of **1** and **2** at 30 °C. **c** The water vapor adsorption and desorption isotherms of **1** and **2** at 30 °C. **d** The Nyquist plots of **2** at 98% RH and various temperatures (30–80 °C).

**e** Arrhenius plot of the proton conductivity of **1** and **2**. **f** The $\nu$(OH) region of water molecules obtained from in situ IR spectra of **1** and **2** under a water vapor pressure of 2.5 kPa ($P/P_0 = 0.90$). The observed bands were reproduced by the sum of four Gaussian peaks. Source data are provided as a Source Data file.

(deuterated water experiment, Supplementary Fig. 29) measurements confirmed that the conduction is based on the transport of protons[56,57]. Moreover, the conductivity value of **2** up to $7.07 \times 10^{-2}$ S cm$^{-1}$ remains nearly two orders of magnitude than that of **1** under the same conditions (70 °C, 98% RH). Therefore, we can reach the conclusion that the enhanced proton conductivity, as well as the lower activation energy of **2**, had been realized by this powerful Ce-mediated precise tailoring strategy from precursor **1**.

The chemical environments of inter- and intra-molecular proton conduction for both compounds had been presented in order to further gain the possible mechanism of enhanced proton conductivity at the molecular level[45,58]. Each {Mo$_{132}$} in **1** is connected to eight adjacent {Mo$_{132}$} by sixteen intermolecular atomic C···O distances around 3.0 Å (Supplementary Fig. 30), while each {Ce$_{11}$Mo$_{96}$} in **2** is connected to seven adjacent {Ce$_{11}$Mo$_{96}$} by twelve intermolecular atomic O···O distances from 2.8 to 3.1 Å (Supplementary Fig. 31), it is worth noting that these hydrogen-bonding interactions could not be formed without the participation of coordinated water on lanthanide Ce ions with Ce 'Outer' and 'On' binding modes. In fact, 4$f$ ions can be utilized to attract water molecules as aqua ligands and to build robust frameworks that can facilitate proton conduction due to their high coordination number and flexible coordination environment[37,46,48,59]. The obvious intermolecular interactions result in both compounds in the 3D continuous array, whose frameworks featuring condensed packing modes are full of crystalline water molecules forming the hydrogen-bonding networks that are beneficial for effective proton hopping and transfer. Besides, the overall energy barrier for proton transport is also determined by the intramolecular proton-exchange process in POMs[58], this theory has been well-established here. In contrast to closed **1**, the half-closed **2** remains better proton conductivity may be due to the following three tips in terms of its intramolecular microenvironment: (1) Open Cavity. It is the most critical factor related to the enhanced proton conductivity. The sealed cavity volume of **1** is *ca.* 905.265 Å$^3$

(Supplementary Fig. 32), and this void has still been retained over 50% with an open motif (*ca.* 487.109 Å$^3$, Supplementary Fig. 33) in **2** after the Ce-mediated molecular tailoring process. (2) Inside {SO$_4^{2-}$} Groups. The un-coordinated O atoms of sulfate groups {$\mu_2$-$\eta^1$:$\eta^1$:$\eta^0$:$\eta^0$ (bidentate)/$\mu_3$-$\eta^1$:$\eta^1$:$\eta^1$:$\eta^0$ (tridentate)} with stronger Lewis acidity may be served as better proton hopping sites than the terminal coordinated O sites of the metal ions. (3) Inside {Ce(H$_2$O)$_x$} ($x = 3$ to 5) Moieties. The utilization of 9-coordinate Ce$^{3+}$ could stabilize the half-closed motif, and its abundant aqua ligands may also participate in building a hydrogen-bonding network inside the cavity. Based on the analysis above, the open and hydrophilic cavity provided by {SO$_4^{2-}$} and {Ce(H$_2$O)$_x$} groups is large enough to trap guest and adsorbed water molecular within **2**, which is confirmed by its higher water uptake capacity in the water vapor adsorption isotherm, realizing dense and extended hydrogen-bonding network (Supplementary Fig. 34, short O···O distances, 2.6–2.9 Å) for promoting proton transport. From this perspective, the states of water molecules in both compounds were explored through in situ infrared spectrum (IR) under water vapor condition[59,60]. The absorption bands around 3000–3700, as well as 1600 cm$^{-1}$, can be assigned to water molecules with OH stretching ($\nu$(OH)) and HOH bending ($\delta$(HOH)), respectively (Supplementary Fig. 35)[59–61]. It has been demonstrated that the band with regard to $\delta$(HOH) of water molecules remains less responsive to hydrogen-bonding networks[62]. More importantly, it is well known that the $\nu$(OH) bands of water molecules in a hydrogen bonding range occur individually, and the positions of the bands reduce as the increase in number and strength of hydrogen bonds[59,60,63]. As expected, the characteristic broad bands appeared gradually after introducing water vapor (Supplementary Fig. 36). Figure 6f highlights the hydroxyl stretching region of the spectra; the presented bands had been deconvoluted into four Gaussian peaks located around (i) 3513–3569 cm$^{-1}$, (ii) 3327–3403 cm$^{-1}$, (iii) 3155–3236 cm$^{-1}$, and (iv) 2988–3012 cm$^{-1}$, which may be classified into the following four groups based on the water molecules in crystal

structures of **1**–**2** as well as previous research about the state of water molecules in Keggin-type and Preyssler-type POM systems reported by Uchida et al.:[59,60] the $\nu$(OH) bands of water molecules (a) in the vicinity of a metal ion (Na$^+$ or Ce$^{3+}$) and without hydrogen-bonds, (b) in the vicinity of a metal ion (Na$^+$ or Ce$^{3+}$) and at a hydrogen-bonding range with an oxide ion or an oxygen atom, (c) in hydrogen-bonding distances with two oxygen atoms and/or oxide ion, and in (c) featuring short hydrogen-bonding distances (d). The IR bands from (i) to (iv) can be attributable to (a)-(d), respectively. Obviously, each band is shifted to lower wavenumbers in **2** compared with **1**, indicating that the hydrogen-bonding network has been particularly strengthened through this powerful Ce-mediated molecular tailoring process.

The proton conductivity stability and durability of both compounds have been fully explored. The cooling and heating conductivity results remain almost unchanged (Supplementary Figs. 37 and 38), indicating their stability under harsh conditions. Furthermore, their proton conductivities were monitored over a monitoring period of 15 h (Supplementary Fig. 39), and little changes were observed. At this level, no obvious changes could be identified in sample morphology of **2** with high proton conductivity through scanning electron microscope (SEM) results (Supplementary Fig. 40). In addition, the stability assessments are further testified by experimental data from the infrared spectrum (IR, Supplementary Fig. 41), Raman (Supplementary Fig. 42), X-ray diffraction (XRD, Supplementary Figs. 43 and 44), Mo$^{V/VI}$ and Ce$^{III}$ XPS (Supplementary Figs. 45–47) of **1** and **2** before and after proton conduction tests. All data suggest that the structural integrity of **1** and **2** are still retained during the tests.

## Discussion

In summary, we have demonstrated a Ce-mediated molecular tailoring strategy from parent **1** to specific product **2** in giant POM assemblies. Both compounds were fully characterized by exhaustive physicochemical techniques in solution and the solid. The tailorable parent features a capped wheel structural motif involving a never seen {Mo$_{110}$} decameric wheel, which also undergoes a quasi-isomerism in 132-nuclearity POMs, namely Mo Brown (spherical Mo$_{132}$) and fresh Mo Green (**1**, capped wheel Mo$_{132}$). To our knowledge, such molecular isomerism phenomenon has not been explored in POMs. **1** possesses high stability in solution, and thus, a Ce-mediated tailoring process was produced by the substituent of edge-shared {Mo$_2$} reactive sites by Ce centers featuring overall 'Inner-On-Outer' binding modes, which drives the resection of one {Mo$_{11}$} cap of **1** by the dissociation of {Mo$_2^*$} clipping sites and finally gives rise to a predictable half-closed product **2**. The proton conductivity results showcase that **2** remains one of the well-known POMs-based crystalline proton conductors that keep high and stable proton conductivity over a relatively wide temperature region. This work constitutes a major step toward the controllable assembly of POM clusters through a precise tailoring process to optimize their whole properties.

## Methods

### Materials and characterization

All reagents and solvents were purchased from commercial sources and used as received. Detailed characterization methods are given in the Supplementary Information.

### Materials synthesis

**1**: Na$_{22}$H$_{30}${Mo$_{132}$O$_{372}$(OH)$_{10}$(H$_2$O)$_{12}$(SO$_4$)$_5$(CH$_3$COO)$_{20}$}·122H$_2$O

Na$_2$S$_2$O$_4$ (0.1 g, 0.60 mmol) and CH$_3$COONa·3H$_2$O (0.1 g, 0.70 mmol) were added to a solution of Na$_2$MoO$_4$·2H$_2$O (0.176 g, 0.72 mmol) in water (12 mL). The clear mixture was then acidified to pH 2.20 with 1 M H$_2$SO$_4$ and stirred for over 30 min. Subsequently, such Mo-containing aqueous solution was transferred to a 25 Teflon-lined stainless-steel container, which was heated to 150 °C and kept for two days, then slowly cooled to room temperature. The filtrate was kept in an open 15 ml beaker for about one month. The dark black crystals were collected and washed with ethanol. Yield: 26 mg (20% based on Mo). IR (cm$^{-1}$): 1620 (s), 1544 (s), 1458 (w), 1413 (s), 980 (s), 858 (m), 812 (s), 750 (w), 649 (m), 572 (s), 477 (w). Elemental analysis % calcd (found): C 2.05 (2.18), H 1.59 (1.47), S 0.69 (0.67), Mo 54.1 (53.5).

**2**: Na$_3$H$_6${Ce$_{11}$Mo$_{96}$O$_{286}$(H$_2$O)$_{101}$(SO$_4$)$_8$}·170H$_2$O

CeCl$_3$ (0.40 g, 1.60 mmol) and Na$_2$SO$_4$ (0.80 g, 5.64 mmol) were added under stirring to a deep green solution of **1** (0.1 g, 0.004 mmol) in H$_2$O (250 mL). The resulting reaction mixture was acidified to pH 2.20 with 2 M H$_2$SO$_4$ and kept at 120 °C for 3 h. After filtration of the hot solution, the deep green filtrate was stored for crystallization. After 1 week, the dark black crystals of **2** were collected by filtration and washed with ethanol. Yield: 29 mg (25% based on Mo). IR (cm$^{-1}$): 1613 (s), 1411 (s), 1127 (w), 976 (s), 873 (w), 816 (s), 724 (w), 649 (m), 558 (s). Elemental analysis % calcd (found): H 2.62 (2.56), S 1.22 (1.35), Mo 43.7 (44.8); Ce 7.32 (7.62).

### Single-crystal X-ray diffraction

Single-crystal X-ray diffraction data for **1** and **2** were recorded on a Bruker Apex CCD II area-detector diffractometer with graphite-monochromated Mo$_{K\alpha}$ radiation ($\lambda = 0.71073$ Å) at 173(2) K. Absorption corrections were applied using multi-scan technique and performed by using the SADABS program. The structures were solved by direct methods and refined on $F^2$ by full-matrix least squares methods by using the SHELXTL minimization on Olex 2.0 software package. Detailed formula determination is given in the Supplementary Information.

### DFT calculations

The single-point calculations were carried out at the (U)B3LYP/6-31G(d)(H, O)/Lanl2DZ(Mo) level. We choose {H$_{42}$Mo$_{16}$O$_{67}$} units in {Mo$_{132}$} ball for theoretical analysis, and in {Mo$_{132}$} capped wheel, we choose both {H$_{11}$Mo$_5$O$_{20}$} units in the wheel and {H$_{62}$Mo$_{26}$O$_{107}$} units that contain reduced {Mo$^V_2$} units for theoretical analysis, the unsaturated oxygen is terminated with H. All DFT calculations were carried out by the Gaussian 16 (Revision D.01) program.

### Water uptake isotherms

The water adsorption–desorption isotherms were measured using a Micrometrics ASAP-2020 Mautomatic specific surface area. After as-synthesized crystalline samples were soaked in C$_2$H$_5$OH for 2 days, the activated process was conducted with the activated temperatures of 423 K under vacuum.

### In situ IR spectra under water vapor condition

In situ IR spectra of both compounds under water vapor were measured on a Bruker Vertex 70 Fourier Transform Infrared Spectrometer. Each powder sample was placed at the center of an IR cell and treated in vacuo at 423 K for 6 h. Then, the powder sample was exposed to water vapor at room temperature with equilibrium pressures of 0.5–2.50 kPa. The equilibrium of the sorption was confirmed by the fact that the spectrum did not change over time. The spectra were deconvoluted with the Gaussian function.

### Proton conductivity

The dried single crystals were ground uniformly into powder. The powder samples were then put into a homemade mold with a radius of 2.50 mm. Pellets were obtained via compression, and their thicknesses were measured using a vernier caliper. Silver glue was spread on both sides of the pellets and allowed to dry in the air. The pellet sample was attached with two copper electrodes. AC impedance measurements were performed on a multichannel electrochemical test system (1470E, Solartron) at a frequency range from 1 MHz to 0.1 Hz. The relative humidity and temperature values were set using a BPS-50CL constant temperature and humidity chamber.

The proton conductivity was calculated using the following equation:

$$\sigma = \frac{L}{SR} \tag{1}$$

where $\sigma$ is the proton conductivity ($S\ cm^{-1}$), $L$ is the thickness of the sample (cm), $S$ is the area of the sample ($cm^2$), and $R$ is the resistance ($\Omega$). The activation energy ($E_a$) was calculated according to the following equation:

$$\sigma T = \sigma_0 e^{\frac{-E_a}{k_B T}} \tag{2}$$

where $T$ is the temperature (K), $\sigma_0$ is the prefactor, and $k_B$ is the Boltzmann constant.

## Data availability

The X-ray crystallographic data for structures reported in this article have been deposited at the Cambridge Crystallographic Data Center (CCDC) under deposition numbers CCDC 2225462 (**1**) and 2225463 (**2**). Both data can be obtained free of charge from The Cambridge Crystallographic Data Center via www.ccdc.cam.ac.uk/data_request/cif. All data supporting the findings of this study are available within the paper, its supplementary information, or the corresponding author. Source data are provided in this paper.

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

## Acknowledgements

This work was financially supported by the NSFC of China (no. 21801038, W.-C.C.), the Natural Science Foundation of Jilin Province—Free Exploration General Project (YDZJ202201ZYTS331, W.-C.C.), and the Science and Technology Research Foundation of Jilin Educational Committee (JJKH20221158KJ, W.-C.C.), the Fundamental Research Funds for the Central Universities (2412022ZD002, W.-C.C.).

## Author contributions

W.-C.C. and X.-L.W. conceived and designed the experiment. X.-X.L. and C.-H.L. conducted synthesis, characterization, and properties. M.-J.H. and C.Q. performed water vapor adsorption tests. W.-C.C, K.-Z.S., and X.-L.W. solved the crystal structures. B.Z. and W.G. conducted density functional theory calculations, W.-C.C., Y.Y., C.-Y.S., X.-L.W., and Z.-M.S. co-wrote the paper. All the authors contributed to the analysis of the results.

## Competing interests

The authors declare no competing interests.
