## [Peer Review File · Nature Communications]

Ce-Mediated Molecular Tailoring on Gigantic Polyoxometalate {Mo₁₃₂} into Half-Closed {Ce₁₁Mo₉₆} for High Proton ConductionREVIEWER COMMENTS

Reviewer #1 (Remarks to the Author):

Wang et al. describes two new types of giant polyoxomolybdates in this manuscript, which I have read with great interest. Three points argue for the significance of this work and its suitability for Nat. Commun.: (1) From structural point of view, the capping of Mo wheels and further addition of Ce ions, transform it into Ce₁₁Mo₉₆, is unusual if not unprecedented. (2) The results showed that hydrothermal reactions could open a whole new world to the chemistry of giant polyoxomolybdates, which had been almost exclusively prepared under ambient conditions. One benefit, for example, could be that crystalline water in the lattice are more regularly arranged, making it more efficient to transport protons through the “hopping” (Grotthuss) mechanism. This also comes to my final point: (3) The exceptional proton conductivity of Ce₁₁Mo₉₆, close to 10⁻¹ S cm⁻¹, is rare and noteworthy, although the reasons laid out by the authors may be subjected to debate (for instance, the very ordered water structures around Ce ions and between molecules, when prepared under hydrothermal conditions, could be a key contributor, I believe). I would be glad to support its publication in Nat. Commun. once following points are properly addressed,

(1) I would recommend additional deuterated water proton conduction experiments to confirm that protons are indeed the charge carriers.

(2) I believe that the color displayed by reduced giant polyoxomolybdates could be related to the degree of Mo reduction (see Angew. Chem. Int. Ed. 2022, 61, e202201672), although the location of reduction could also a factor as the authors claimed. But the statement “ When the reducing electrons are delocalized not only over the ring but also the isolated Mo–Mo bonds in the {Mo(V)₂} units” are not correct as stated. The electrons are delocalized among Mo sites, meaning the Mo sites are reduced Mo(V) in some molecules but not others. However, the electrons are not delocalized over bonds.

(3) The thermal stability of the materials should be better supported by experimental evidence. For instance, the activated temperatures of the materials related to the water vapor adsorption analysis should be given...

(4) In the experimental section, essential parameters for proton conductivity measurements should be provided in a separate section: sample effective area, thickness, frequency and temperature range, etc.

(5) I would suggest the authors to break down some of the paragraphs into shorter ones whenever possible & appropriate. Long paragraphs are really hard to follow.

(6) Page 7, line 191, “we were able to get half-closed Ce-containing product 2 after a careful optimization of the reaction conditions”, what parameters/conditions were optimized?.

(7) Page 8, line 217, the statement: “Ce Inner-Outer binding modes are defined as a set of two Ce centers take the place of {Mo₂} units both inner and outer surfaces of major molybdenum framework.” is confusing to me. Please clarify.

(8) Page 10, line 272, the authors stated that the bulk conductivity of Ce₁₁Mo₉₆ is less temperature dependent over a relatively wide range of temperature. How about its proton conductivity below 273 K?

(9) Page 3, Line 214-215, “which produces an asymmetric molecule tailoring process that has not been established in other POMs” looks like an overstatement. There are some known and established methods to tailor/design asymmetric POM molecules; introducing chiral functional groups is apparently a common one...

(10) The manuscript may benefit from some English polishing and grammatical checks. E.g. Page 2, line 27, Change “aroused much concern” to maybe “attracted much interest” as the former has a negative connotation and that’s not what the authors meant. Page 2, line 51, “importance” not “important”. There are others...

Reviewer #2 (Remarks to the Author):

The controllable preparation of giant POM clusters including precise atomic-level tailoring and structural functionality remains a great challenge. In this paper, the authors presented the first case of Ce-mediated molecular tailoring on the unprecedented gigantic {Mo₁₃₂} into half-closed product {Ce₁₁Mo₉₆}. The {Mo₁₃₂} capped wheel undergoes a rare quasi-isomerism with Keplerate {Mo₁₃₂} ball and half-closed {Ce₁₁Mo₉₆} achieves high proton conduction, nearly two orders of magnitude than that of {Mo₁₃₂} capped wheel. This is a pretty good article outlining the controllable synthesis and enhanced structural functionality of POMs. Minor revisions are suggested as follows.

1. The paper is lacking an independent analysis part as this one seems to be included in the synthesis discussion from {Mo₁₃₂} to {Ce₁₁Mo₉₆}, which could help to enrich the interesting molecular tailoring process.
2. Are there any explanations about the really noisy XPS spectra presented here for Ce if we compare them to those for the Mo atoms? A short explanation could be added.
3. A figure including the comparison of Mo-based building blocks and their connection rules in Mo₁₃₂ ball and Mo₁₃₂ capped wheel should be given, it is beneficial for highlighting the quasi-isomers in 132-nuclearity polyoxometalates.
4. It is better for the authors to provide the experimental and simulated PXRDs, these should be clearly mentioned for crystallinity and purity of the samples.
5. The authors mentioned that ‘Ce Inner and Inner-Outer binding modes had not been reported in 4f-doped gigantic Mo Wheels’, to my knowledge, Ce Inner binding modes had already been discovered in {Mo₁₀₀Ce₆}.
6. In Figure 4c, check the made-up word ‘inner-outer’.

7. References: several references lack final pages and show inappropriate format, please check them.

8. To benefit a broad readership, some related works about POMs should be added in the introduction part, such as *Angew. Chem. Int. Ed.* 60, 6382–6385(2021); *Polyoxometalates 1*, 9140006(2022); *Polyoxometalates 1*, 9140011(2022); *Polyoxometalates 2*, 9140020(2023).

Reviewer #3 (Remarks to the Author):

The manuscript demonstrated an unparalleled Ce-mediated molecular tailoring strategy from a closed {Mo₁₃₂} capped wheel to specific product half-closed product {Ce₁₁Mo₉₆} without altering the major framework. This is always a highly desirable strategy for customizing functional materials through precisely atomic-level tailoring and structural functionality. By this directional tailoring, the target {Ce₁₁Mo₉₆} achieves high proton conduction, which is nearly two orders of magnitude than that of parent {Mo₁₃₂}. This work is interesting and of broad interest to the material and chemistry community; thorough characterization is provided for each step and mechanism. However, the following questions/comments should be addressed prior to the publication of the manuscript:

1. As the new structure, PXRD compared to simulated patterns of the two compounds should be provided.

2. The synthetic procedures and characterization methods should be described more detailly, such as the color changes of each step and the detailed process of proton conductivity measurements, so that the work can better be reproduced by future researchers.

3. The compound 2 is tailored from parent compound 1, but its yield is low to 16 %. What happens to the other part of parent 1 during this functionalization process?

4. The TGA curve of compound 1 rises slowly during the 450 °C to 750 °C, why? What reaction might have undergone? In addition, for easier comparison, the TGA curves for 1 and 2 should be in the same range of horizontal and vertical coordinates.

5. Ce is used here as a model mediator. To be able to use other metal ions, what are the suitable characteristics of the metal ions?

6. The Figure format of the full text should be uniform, for example, the format of Figures S15 to S19 is not consistent with other Figures. In addition, the color of the Raw in XPS is so light that it is almost impossible to see if the fitted data overlaps well with them. The fitted XPS spectra of Ce in Figure S19 and Figure S37 is not matched well with the Raw data, whether this will affect the valence state determination of Ce, please check it.

7. To improve readability, the band distance of Mo···Mo can be labelled on the Figure 1e and Figure 1g.
8. Does the two isomerism {Mo₁₃₂} have identical reduction degree (45%, Mo^V60Mo^{VI}72)? How did the author calculate and get this conclusion?
9. The delocalization of the reducing electrons is the key for different optical absorptions of POMs. Authors attribute this new 'Mo Green' to delocalize reducing electrons not only over the ring but also the isolated Mo–Mo bonds in {MoV₂} units, how did the author get this conclusion?
10. Please provide the electrical equivalent circuit that used for fitting the Nyquist plot and elaborate on the calculation process of resistance.

Reviewer #4 (Remarks to the Author):

The discovery of the Mo₁₃₂ capped wheel (1), which is “quasi-isomeric” with the previously reported Mo₁₃₂ ball, is noteworthy in the research area of giant Mo-oxo clusters. In particular, 1 contains Mo₂ reactive sites, which can be eliminated by Ce coordination to produce Ce₁₁Mo₉₆ (2) with an open cavity. The open cavity of Ce₁₁Mo₉₆ (2) leads to high water adsorption property and proton conductivity, which are comparable to those of previously reported POM-based or MOF-based proton conductors. These findings would potentially interest the readers of Nature Communications. However, the crystal structure of the bulk powder of 1 and 2, which is crucial for assessing the optical properties and proton conductivities, is missing. Additionally, most of the discussions on the optical properties and proton conductivities are based on the crystal structure, without any supporting spectroscopic measurements or theoretical calculations. Therefore, major revision is required to meet the standards of publication in Nature communications.

(1) Please provide the CHNS analysis of compounds 1 and 2.

(2) Please provide more details on the choice of synthetic conditions, such as the pH and temperature. Why are the yields of 1 and 2 rather low (16% and 20% based on Mo)? If you could only obtain 26 and 5 mg of 1 and 2, respectively, from one batch, it would be quite difficult to carry out characterization and function evaluation.

(3) Why did the authors focus on Ce among the lanthanides to dissociate the Mo ball?

(4) There are two 132-Mo-atoms clusters in this paper; Mo₁₃₂ ball and Mo₁₃₂ capped wheel. Please provide the definition of “quasi-isomers” and “quasi-isomerism”. Please also compare the synthetic conditions of the isomers.

(5) The UV-vis spectra and Moss-Schottky measurements show that the Mo₁₃₂ ball and the Mo₁₃₂ capped wheel show different optical behaviors, which the authors attribute to “the difference in

delocalization of the reducing electrons". Could you please provide experimental or theoretical data (such as electronic band structure calculations) to support this hypothesis?

(6) Bulk powder PXRD data for 1 and 2 should be shown and compared with simulated patterns obtained from single crystal data.

(7) The structural stability after the proton conduction measurements is evaluated by IR and Raman spectra. However, these data can only provide information on the molecular structure. Therefore, PXRD data after the proton conductivity measurement are necessary to assess the structural stability of the compounds under the measurement conditions.

(8) Please show the equivalent circuit for fitting the Nyquist plot.

(9) Readers may find it interesting if the "quasi-isomer effect" of the 132-Mo-atoms clusters were presented.

(10) The authors' statement that "the dense and extended hydrogen-bonding network of 2 promotes proton transport" is inconsistent with the higher activation energy for proton conduction in 2 than in 1.

(11) Related with (10), the proton conduction mechanism should be supported by spectroscopic evidence, rather than solely based on the crystal structure. Please consider in-situ IR measurements under water vapor (ACS Applied Mater Interfaces 2021, 13, 19138), DQF 1H NMR (JACS 2016, 138, 27, 8505; JACS 2021 143, 21433), or neutron scattering techniques (J. Phys. Chem. Lett. 2018, 9, 5772).

(12) The effect of lanthanide ions on proton conduction has been explored with POM-based compounds. Please refer to ACS Applied Mater Interfaces 2021, 13, 19138.

Reviewer #1 (Remarks to the Author)

Wang et al. describes two new types of giant polyoxomolybdates in this manuscript, which I have read with great interest. Three points argue for the significance of this work and its suitability for *Nat. Commun.*: (1) From structural point of view, the capping of Mo wheels and further addition of Ce ions, transform it into Ce₁₁Mo₉₆, is unusual if not unprecedented. (2) The results showed that hydrothermal reactions could open a whole new world to the chemistry of giant polyoxomolybdates, which had been almost exclusively prepared under ambient conditions. One benefit, for example, could be that crystalline water in the lattice are more regularly arranged, making it more efficient to transport protons through the “hopping” (Grotthuss) mechanism. This also comes to my final point: (3) The exceptional proton conductivity of Ce₁₁Mo₉₆, close to 10⁻¹ S cm⁻¹, is rare and noteworthy, although the reasons laid out by the authors may be subjected to debate (for instance, the very ordered water structures around Ce ions and between molecules, when prepared under hydrothermal conditions, could be a key contributor, I believe). I would be glad to support its publication in *Nat. Commun.* once following points are properly addressed.

Question 1: I would recommend additional deuterated water proton conduction experiments to confirm that protons are indeed the charge carriers.

Answer: Thank you for your helpful suggestions. {Ce₁₁Mo₉₆} exhibits higher proton conduction than that of {Mo₁₃₂}, so we chose {Ce₁₁Mo₉₆} to explore the deuterated water proton conduction experiment, see *Nat. Mater.* **15**, 722–726(2016), *Chem. Sci.* **10**, 556–563(2019), *Inorg. Chem. Front.* **5**, 1213-1217(2018). The dehydration {Ce₁₁Mo₉₆} is placed to saturated salt solution of NH₄Cl (~80% RH) for 24 hours at 303 K. Impedance analysis of deuterated samples exhibited proton conductivity value of 2.02×10⁻⁴ S cm⁻¹ for {Ce₁₁Mo₉₆} at 303 K (as shown in Figure S28 in the supporting information). The decrease in proton conductivity is due to the isotopic effect of deuterium. Hence, the proportion follows the equality: $\sigma(\text{H}) : \sigma(\text{D}) = 1.40$ for {Ce₁₁Mo₉₆}, where H, D represent the hydrogen and deuterium, see *J. Mater. Chem. A.* **5**, 7243-7256(2017). Related discussions had already been included in the main text and Supporting Information.

Question 2: I believe that the color displayed by reduced giant polyoxomolybdates could be related to the degree of Mo reduction (see *Angew. Chem. Int. Ed.* **2022**, *61*, e202201672), although the location of reduction could also a factor as the authors claimed. But the statement “When the reducing electrons are delocalized not only over the ring but also the isolated Mo–Mo bonds in the {Mo(V)₂} units” are not correct as stated. The electrons are delocalized among Mo sites, meaning the Mo sites are reduced Mo(V) in some molecules but not others. However, the electrons are not delocalized over bonds.

Answer: Thank you for your helpful corrections. The statement “but also the isolated Mo–Mo bonds in the {Mo(V)₂} units” is based on the statement in Cronin's article, see *Bull. Jpn. Soc. Coord. Chem.* **78**, 11-17(2021), the statement is “This is because the electrons are localized in reduced and isolated {Mo^V}₂ Mo–Mo bonds”. Thank you for your correction, we also thought the statement in our paper is inappropriate, this part had been revised as follow “.....When the reducing electrons are delocalized not only over the ring but also the {Mo^V}₂ units.....”.

Question 3: The thermal stability of the materials should be better supported by experimental evidence. For instance, the activated temperatures of the materials related to the water vapor adsorption analysis should be given.

Answer: Thank you for your helpful suggestions. The activated temperatures of the materials related to the water vapor adsorption were determined based on TGA analysis. TGA results show that {Mo₁₃₂} and {Ce₁₁Mo₉₆} possess thermal stabilities at about 150°C. The activated temperatures of the materials had been given as follows: “The water adsorption–desorption isotherms were measured using a Micrometrics ASAP-2020 Mautomatic specific surface area. After both crystalline samples were soaked in C₂H₅OH for 2 days, the activated process was conducted with the activated temperatures of 423 K under vacuum”.

Question 4: In the experimental section, essential parameters for proton conductivity measurements should be provided in a separate section: sample effective area, thickness, frequency and temperature range, etc.

Answer: Thank you for your helpful suggestions. The essential parameters for proton conductivity measurements had been provided in a separate section in the supporting information as follows:

The dried single crystals were ground uniformly into powder. The powder samples were then put into a homemade mold with a radius of 2.50 mm. Pellets were obtained *via* compression and their thicknesses were measured using a vernier caliper. Silver glue was spread on both sides of the pellets and allowed to dry in air. The pellet sample was attached with two copper electrodes. Alternating-Current (AC) impedance measurements were performed on a multichannel electrochemical test system (1470E, Solartron) at a frequency range from 1 MHz to 0.1 Hz. The relative humidity and temperature values were set using a BPS-50CL constant temperature and humidity chamber. The bulk proton conductivity was calculated by fitting the Nyquist plot with an electrical equivalent circuit below, which consists of a contact resistor (R1), bulk resistor (R2), capacitance (CPE1), and Warburg diffusion element (W1).

The proton conductivity was calculated using the following equation:

$$\sigma = \frac{L}{SR}$$

where σ is the proton conductivity ($S \cdot cm^{-1}$), L is the thickness of the sample (cm), S is the area of the sample (cm^2), and R is the resistance (Ω). The activation energy (E_a) was calculated according to the following equation:

$$\sigma T = \sigma_0 e^{\frac{-E_a}{k_B T}}$$

where T is the temperature (K), σ_0 is the prefactor, and k_B is the Boltzmann constant.

Question 5: I would suggest the authors to break down some of the paragraphs into shorter ones whenever possible & appropriate. Long paragraphs are really hard to follow.

Answer: Thank you for your helpful suggestions. Some of the paragraphs had been simplified for shorter ones, and now the paper is much easier to follow.

Question 6: Page 7, line 191, “we were able to get half-closed Ce-containing product 2 after a careful optimization of the reaction conditions”, what parameters/conditions were optimized?

Answer: Thank you for your helpful suggestions. An independent synthesis discussion from {Mo₁₃₂} to {Ce₁₁Mo₉₆} had been added into supporting information due to length limitations in main text, including the parameters/conditions during the optimization. The synthesis discussion is shown as follows.

Synthesis discussion of 2: During the synthesis, two important factors should be emphasized: (1) the choice of Ce ions. 4f lanthanide (Ln) ions are oxophilic and display high coordination numbers as well as relatively long Ln-O bond lengths, which are usually used as strong electrophiles to replace edge-sharing or corner-sharing {Mo₂} units on the parent polyoxomolybdate framework. For example, Ln^{III} center with the coordination configuration of a distorted monocapped square antiprism, built from four μ_2 -O atoms and five H₂O molecules – {Ln(H₂O)₅}, joins two sets of {Mo₆} with four Mo–O–Ce bonds, see *J. Am. Chem. Soc.* **136**, 14114–14120(2014), in a manner similar to a {Mo₂} unit (both act as 4-connected rods). Such coordination configuration remains the key point why lanthanide ions are easy to be incorporated into polyoxomolybdate wheels. In all, we choose lanthanide ions to dissociate the Mo ball mainly due to its high coordination number (nine) and flexible coordination environment ({Ln(H₂O)₅, distorted monocapped square antiprism). In fact, during the preparation of {Ce₁₁Mo₉₆}, all 4f lanthanide ions (except radioelement) were introduced into the reaction system for exploring the Ln-mediated molecule tailoring process. When lanthanide ion was absent, no crystals but precursor {Mo₁₃₂} was obtained. Simple 3d metal salts (*e.g.* Co²⁺, Mn²⁺, Fe³⁺, *etc.*) or metal coordination complexes (*e.g.* {Mn₁₂}, Ni complex [Ni₂(μ -OH₂)(O₂CCMe₃)₄(HO₂CCMe₃)₄], Cr complex [Cr₃O(OOCCH₂CN)₆(H₂O)₃]⁺, *etc.*) were

also introduced to the acid solution, but no crystals were obtained. We found that only 4f lanthanide La, Ce and Pr ions could be harvested with crystal samples. However, La- or Pr-containing samples possess rather poor crystal quality that the data are really hard to be collected although careful optimization related to the reaction conditions had been conducted. So we just report the Ce-containing samples with good crystal quality for publication. At this level, previous reports show that the Ln-containing isomers in POMs possess similar IR spectra for polyoxometalate “fingerprint region”, see *Dalton Trans.* 4423–4425(2009), *Chem. Eur. J.* **20**, 12144–12156(2014), *Chem. Eur. J.* **21**, 18168–18176(2015). As shown in Figure 1.1, the current La-, Ce- and Pr-containing samples possess similar IR spectra for polyoxometalate “fingerprint region”. So we guess that Ln-mediated molecule tailoring process could be realized through La, Ce and Pr ions. The lanthanide contraction results in smaller ionic radii as we cross the 4f period which becomes a key factor in the replacement of {Mo₂} from Pr onward. This phenomenon is common in Ln-containing giant Mo-oxo clusters, see Table 1.1. We found that the 4f lanthanide ions are mainly focus on early lanthanide ions or just Ce ions.

Figure 1.1 IR spectra for polyoxometalate “fingerprint region” of La-, Ce- or Pr-containing samples.

Table 1.1 Some reported representative Ln-containing giant Mo-oxo clusters.

Year	Compounds	lanthanide ions	Ref.
2022	$[\text{Mo}_{64}\text{Ni}_8\text{Ln}_6\text{H}_{26}\text{O}_{200}(\text{H}_2\text{O})_{30}]^{8-}$	La, Ce, Pr	Angew. Chem. Int. Ed. e202201672 (2022)
2021	$[\text{Mo}_{124}\text{Ce}_4\text{O}_{376}(\text{H}_2\text{O})_{64}\text{H}_{12}(\text{C}_8\text{H}_{13}\text{N}_4\text{O}_3)_4]^{3-}$	Ce	Chem. Sci. 12 , 2427– 2432(2021)
2020	$[\text{H}_{16}\text{Mo}_{124}\text{Ce}_4\text{O}_{376}(\text{H}_2\text{O})_{56}(\text{PMo}_{12}\text{O}_{40})(\text{C}_6\text{H}_{12}\text{N}_2\text{O}_4\text{S}_2)_4]^{5-}$	Ce	Proc. Natl. Acad. Sci 117 , 10699–10705(2020)
2019	$\{(\text{Mo}_8\text{O}_{26})\text{Mo}_{124}\text{Ce}_4\text{O}_{376}(\text{H}_2\text{O})_{60}\text{H}_{12}(\text{C}_6\text{H}_9\text{N}_3\text{O}_2)_6\}^{12-}$ $\{\text{Mo}_{124}\text{Ce}_4\text{O}_{376}(\text{H}_2\text{O})_{60}\text{H}_{12}(\text{C}_6\text{H}_{14}\text{N}_4\text{O}_2)_6\}^{8-}$	Ce	J. Am. Chem. Soc. 141 , 1242–1250(2019)
2019	$[\text{H}_{12}\text{Mo}_{128}\text{Ce}_4\text{O}_{389}(\text{H}_2\text{O})_{60}(\text{C}_5\text{H}_{13}\text{N}_2\text{O}_2)_6]^{4-}$ $[\text{H}_{14}\text{Mo}_{158.5}\text{Ce}_2\text{O}_{478}(\text{H}_2\text{O})_{81}(\text{C}_5\text{H}_{13}\text{N}_2\text{O}_2)_6]^{5-}$	Ce	Angew. Chem. Int. Ed. 58 , 10867–10872(2019)
2017	$\text{Ce}_{0.5}[\text{Ce}_6\text{H}_{16.5}\text{Mo}_{130}\text{O}_{396}(\text{H}_2\text{O})_{84}]$	Ce	Angew. Chem. Int. Ed. 56 , 9727–9731(2017)
2014	$[\text{Mo}_{100}\text{Ce}_6\text{O}_{306}\text{H}_{10}(\text{H}_2\text{O})_{71}]_2^{8-}$ $[\text{Mo}_{100}\text{Ce}_6\text{O}_{306}\text{H}_{10}(\text{H}_2\text{O})_{70}]^{4-}$	Ce	J. Am. Chem. Soc. 136 , 14114–14120(2014)
2010	$[\text{Mo}_{96}\text{O}_{301}(\text{H}_2\text{O})_{29}\{\text{La}(\text{H}_2\text{O})_6\}_2]\{\text{La}(\text{H}_2\text{O})_5\}_6]^{22-}$	La	Inorg. Chem. 49 , 9426– 9437(2010)
2006	$[\text{Mo}_{150}\text{O}_{452}\text{H}_2(\text{H}_2\text{O})_{66}\{\text{La}(\text{H}_2\text{O})_5\}_2]^{24-}$	La	J. Alloys Compd. 408 , 693– 700(2006)
2002	$\{[\text{Mo}_{128}\text{Eu}_4\text{O}_{388}\text{H}_{10}(\text{H}_2\text{O})_{81}]_2\}^{20-}$	Eu	Angew. Chem. Int. Ed. 41 , 2805-2808(2002)
2000	$[\text{Mo}_{120}\text{O}_{366}(\text{H}_2\text{O})_{48}\text{H}_{12}\{\text{Pr}(\text{H}_2\text{O})_5\}_6]^{6-}$	Pr	Inorg. Chem. 39 , 3112- 3113(2000)

(2) the use of additional SO_4^{2-} . When additional Na_2SO_4 was introduced into this reaction system, the $\{\text{Ce}_{11}\text{Mo}_{96}\}$ product could be obtained with a relatively high yield (~25%). The yield without Na_2SO_4 is 16%. Comparative experiments showed that the yield keeps unchanged when Na_2SO_4 was replaced by NaCl or $\text{CH}_3\text{COONa}\cdot 3\text{H}_2\text{O}$, which we found that the ionic strength of sulfite anion in the reaction solution possesses certain influence on the synthesis. In general, the sulfite anion as inorganic ligand introduces the necessary diversity into POM systems, see *Inorg. Chem.* **53**, 9486–9497(2014); *Angew. Chem. Int. Ed.* **42**, 2085–2090(2003);

Angew. Chem. Int. Ed. **47**, 8420–842(2008). This is in line with the structural analysis, the SO_4^{2-} are important factors during the assembly of $\{\text{Ce}_{11}\text{Mo}_{96}\}$. Sulfite anion inside half-closed $\{\text{Ce}_{11}\text{Mo}_{96}\}$ provides a suitable microenvironment for trapping ‘inner’ cerium centers, which plays a key role in the formation of Ce-containing product $\{\text{Ce}_{11}\text{Mo}_{96}\}$.

Question 7: Page 8, line 217, the statement: “Ce Inner-Outer binding modes are defined as a set of two Ce centers take the place of $\{\text{Mo}_2\}$ units both inner and outer surfaces of major molybdenum framework.” is confusing to me. Please clarify.

Answer: Thank you for your helpful corrections. This part has been revised as “Ce Inner-Outer binding modes are defined as a set of two Ce centers that are located at both inner and outer surfaces of major molybdenum framework take the place of $\{\text{Mo}_2\}$ units”.

Question 8: Page 10, line 272, the authors stated that the bulk conductivity of $\text{Ce}_{11}\text{Mo}_{96}$ is less temperature dependent over a relatively wide range of temperature. How about its proton conductivity below 273 K?

Answer: Thank you for your helpful questions. Most proton-conducting materials display poor conductivity at subzero temperatures, which significantly limits their application in cold regions. Thus, the exploration of proton-conducting materials with high proton conductivity from subzero to medium temperatures is a significant topic, see *Angew. Chem. Int. Ed.* **57**, 8416–8420(2018). We conducted proton conductivity experiment of $\{\text{Ce}_{11}\text{Mo}_{96}\}$ when the temperature is 263 K (below 273 K), a low proton conductivity ($< 10^{-7} \text{ S cm}^{-1}$) is observed. The conductivity of the hydrated proton-conducting material will drop greatly below 0 °C, see *J. Am. Chem. Soc.* **137**, 913–918(2015), which is common as most proton-conducting materials also display poor proton conductivity at subzero temperature due to the solidification (freezing) of water, see *J. Am. Chem. Soc.* **136**, 13166–13169(2014). As presented in this work, the proton conductivity of $\{\text{Ce}_{11}\text{Mo}_{96}\}$ is highly humidity dependent. Only when the relative humidity is 98%, the bulk conductivity of $\{\text{Ce}_{11}\text{Mo}_{96}\}$ is less temperature dependent over a relatively wide range of temperature. The high relative humidity could not be kept when the temperature is

below 273 K (that is anhydrous environment). In the future, we will pay more attention to build a new strategy to design proton-conducting materials applied in cold climates.

Question 9: Page 3, Line 214-215, “which produces an asymmetric molecule tailoring process that has not been established in other POMs” looks like an overstatement. There are some known and established methods to tailor/design asymmetric POM molecules; introducing chiral functional groups is apparently a common one...

Answer: Thank you for your helpful corrections. This statement had been deleted.

Question 10: The manuscript may benefit from some English polishing and grammatical checks. E.g. Page 2, line 27, Change “aroused much concern” to maybe “attracted much interest” as the former has a negative connotation and that’s not what the authors meant. Page 2, line 51, “importance” not “important”. There are others...

Answer: Thank you for your helpful suggestions. The manuscript had been polished by native speakers for some English polishing and grammatical checks, and now it is more fluency for reading.

Thanks a lot for the referee’s kindly comments!

Reviewer #2 (Remarks to the Author)

The controllable preparation of giant POM clusters including precise atomic-level tailoring and structural functionality remains a great challenge. In this paper, the authors presented the first case of Ce-mediated molecular tailoring on the unprecedented gigantic $\{\text{Mo}_{132}\}$ into half-closed product $\{\text{Ce}_{11}\text{Mo}_{96}\}$. The $\{\text{Mo}_{132}\}$ capped wheel undergoes a rare quasi-isomerism with Keplerate $\{\text{Mo}_{132}\}$ ball and half-closed $\{\text{Ce}_{11}\text{Mo}_{96}\}$ achieves high proton conduction, nearly two orders of magnitude than that of $\{\text{Mo}_{132}\}$ capped wheel. This is a pretty good article outlining the controllable synthesis and enhanced structural functionality of POMs. Minor revisions are suggested as follows.

Question 1: The paper is lacking an independent analysis part as this one seems to be included in the synthesis discussion from $\{\text{Mo}_{132}\}$ to $\{\text{Ce}_{11}\text{Mo}_{96}\}$, which could help to enrich the interesting molecular tailoring process.

Answer: Thank you for your helpful suggestions. An independent synthesis discussion from $\{\text{Mo}_{132}\}$ to $\{\text{Ce}_{11}\text{Mo}_{96}\}$ had been added into supporting information due to length limitations in main text. The synthesis discussion is shown as follows.

Figure 2.1 IR spectra for polyoxometalate “fingerprint region” of La-, Ce- or Pr-containing samples.

Synthesis discussion of 2: During the synthesis, two important factors should be emphasized: (1) the choice of Ce ions. 4f lanthanide (Ln) ions are oxophilic and display high coordination numbers as well as relatively long Ln-O bond lengths, which are usually used as strong electrophiles to replace edge-sharing or corner-sharing {Mo₂} units on the parent polyoxomolybdate framework. For example, Ln^{III} center with the coordination configuration of a distorted monocapped square antiprism, built from four μ_2 -O atoms and five H₂O molecules – {Ln(H₂O)₅}, joins two sets of {Mo₆} with four Mo–O–Ce bonds, see *J. Am. Chem. Soc.* **136**, 14114–14120(2014), in a manner similar to a {Mo₂} unit (both act as 4-connected rods). Such coordination configuration remains the key point why lanthanide ions are easy to be incorporated into polyoxomolybdate wheels. In all, we choose lanthanide ions to dissociate the Mo ball mainly due to its high coordination number (nine) and flexible coordination environment ({Ln(H₂O)₅, distorted monocapped square antiprism). In fact, during the preparation of {Ce₁₁Mo₉₆}, all 4f lanthanide ions (except radioelement) were introduced into the reaction system for exploring the Ln-mediated molecule tailoring process. When lanthanide ion was absent, no crystals but precursor {Mo₁₃₂} was obtained. Simple 3d metal salts (e.g. Co²⁺, Mn²⁺, Fe³⁺, etc.) or metal coordination complexes (e.g. {Mn₁₂}, Ni complex [Ni₂(μ -OH₂)(O₂CCMe₃)₄(HO₂CCMe₃)₄], Cr complex [Cr₃O(OOCCH₂CN)₆(H₂O)₃]⁺, etc.) were also introduced to the acid solution, but no crystals was obtained. We found that only 4f lanthanide La, Ce and Pr ions could be harvested with crystal samples. However, La- or Pr-containing samples possess rather poor crystal quality that the data are really hard to be collected although careful optimization related to the reaction conditions had been conducted. So we just report the Ce-containing samples with good crystal quality for publication. At this level, previous reports show that the Ln-containing isomers in POMs possess similar IR spectra for polyoxometalate “fingerprint region”, see *Dalton Trans.* 4423–4425(2009), *Chem. Eur. J.* **20**, 12144–12156(2014), *Chem. Eur. J.* **21**, 18168–18176(2015). As show in Figure 2.1, the current La-, Ce- and Pr-containing samples possess similar IR spectra for polyoxometalate “fingerprint region”. So we guess that Ln-mediated molecule tailoring process could be realized through La, Ce and Pr ions. The lanthanide contraction results in smaller ionic radii as we cross the 4f period which becomes a

key factor in the replacement of {Mo₂} from Pr onward. This phenomenon is common in Ln-containing giant Mo-oxo clusters, see Table 2.1. We found that the 4f lanthanide ions are mainly focus on early lanthanide ions or just Ce ions.

Table 2.1 Some reported representative Ln-containing giant Mo-oxo clusters.

Year	Compounds	lanthanide ions	Ref.
2022	[Mo ₆₄ Ni ₈ Ln ₆ H ₂₆ O ₂₀₀ (H ₂ O) ₃₀] ⁸⁻	La, Ce, Pr	Angew. Chem. Int. Ed. e202201672 (2022)
2021	[Mo ₁₂₄ Ce ₄ O ₃₇₆ (H ₂ O) ₆₄ H ₁₂ (C ₈ H ₁₃ N ₄ O ₃) ₄] ³⁻	Ce	Chem. Sci. 12 , 2427–2432(2021)
2020	[H ₁₆ Mo ₁₂₄ Ce ₄ O ₃₇₆ (H ₂ O) ₅₆ (PMo ₁₂ O ₄₀)(C ₆ H ₁₂ N ₂ O ₄ S ₂) ₄] ⁵⁻	Ce	Proc. Natl. Acad. Sci 117 , 10699–10705(2020)
2019	{(Mo ₈ O ₂₆)Mo ₁₂₄ Ce ₄ O ₃₇₆ (H ₂ O) ₆₀ H ₁₂ (C ₆ H ₉ N ₃ O ₂) ₆ } ¹²⁻ {Mo ₁₂₄ Ce ₄ O ₃₇₆ (H ₂ O) ₆₀ H ₁₂ (C ₆ H ₁₄ N ₄ O ₂) ₆ } ⁸⁻	Ce	J. Am. Chem. Soc. 141 , 1242–1250(2019)
2019	[H ₁₂ Mo ₁₂₈ Ce ₄ O ₃₈₉ (H ₂ O) ₆₀ (C ₅ H ₁₃ N ₂ O ₂) ₆] ⁴⁻ [H ₁₄ Mo _{158.5} Ce ₂ O ₄₇₈ (H ₂ O) ₈₁ (C ₅ H ₁₃ N ₂ O ₂) ₆] ⁵⁻	Ce	Angew. Chem. Int. Ed. 58 , 10867–10872(2019)
2017	Ce _{0.5} [Ce ₆ H _{16.5} Mo ₁₃₀ O ₃₉₆ (H ₂ O) ₈₄]	Ce	Angew. Chem. Int. Ed. 56 , 9727–9731(2017)
2014	[Mo ₁₀₀ Ce ₆ O ₃₀₆ H ₁₀ (H ₂ O) ₇₁] ₂ ⁸⁻ [Mo ₁₀₀ Ce ₆ O ₃₀₆ H ₁₀ (H ₂ O) ₇₀] ₄ ⁴⁻	Ce	J. Am. Chem. Soc. 136 , 14114–14120(2014)
2010	[Mo ₉₆ O ₃₀₁ (H ₂ O) ₂₉ {La(H ₂ O) ₆] ₂ {La(H ₂ O) ₅] ₆] ²²⁻	La	Inorg. Chem. 49 , 9426–9437(2010)
2006	[Mo ₁₅₀ O ₄₅₂ H ₂ (H ₂ O) ₆₆ {La(H ₂ O) ₅] ₂] ²⁴⁻	La	J. Alloys Compd. 408 , 693–700(2006)
2002	[{Mo ₁₂₈ Eu ₄ O ₃₈₈ H ₁₀ (H ₂ O) ₈₁] ₂] ²⁰⁻	Eu	Angew. Chem. Int. Ed. 41 , 2805–2808(2002)
2000	[Mo ₁₂₀ O ₃₆₆ (H ₂ O) ₄₈ H ₁₂ {Pr(H ₂ O) ₅] ₆] ⁶⁻	Pr	Inorg. Chem. 39 , 3112–3113(2000)

(2) the use of additional SO₄²⁻. When additional Na₂SO₄ was introduced into this reaction system, the {Ce₁₁Mo₉₆} product could be obtained with a relatively high yield (~25%). The yield without Na₂SO₄ is 16%. Comparative experiments showed that the yield keeps unchanged when Na₂SO₄ was replaced by NaCl or

CH₃COONa·3H₂O, which we found that the ionic strength of sulfite anion in the reaction solution possesses certain influence on the synthesis. In general, the sulfite anion as inorganic ligand introduces the necessary diversity into POM systems, see *Inorg. Chem.* **53**, 9486–9497(2014); *Angew. Chem. Int. Ed.* **42**, 2085–2090(2003); *Angew. Chem. Int. Ed.* **47**, 8420–842(2008). This is in line with the structural analysis, the SO₄²⁻ are important factors during the assembly of {Ce₁₁Mo₉₆}. Sulfite anion inside half-closed {Ce₁₁Mo₉₆} provides a suitable microenvironment for trapping ‘inner’ cerium centers, which plays a key role in the formation of Ce-containing product {Ce₁₁Mo₉₆}.

Question 2: Are there any explanations about the really noisy XPS spectra presented here for Ce if we compare them to those for the Mo atoms? A short explanation could be added.

Answer: Thank you for your helpful suggestions. The XPS data for cerium is really noisy presented here, but the given XPS data of Mo atoms remain less noisy. This may be resolution though this following reason: The XPS Ce 3*d* spectra of cerium compounds are known to be complicated due to hybridization of the Ce 4*f* with ligand orbitals and fractional occupancy of the valence 4*f* orbitals. see *Langmuir* **12**, 1794–1799(1996); *J. Phys. Chem. C* **117**, 9520–9528(2013); *J. Phys. Chem. C* **116**, 19182–19190(2012), etc. Based on the above considerations, a short explanation had been added in the X-ray photoelectron spectroscopy (XPS) part in supporting information: It is noting that the Ce 3*d* XPS spectra of Ce-containing compounds are known to be complicated due to the hybridization of the Ce 4*f* with ligand orbitals and fractional occupancy of the 4*f* valence orbitals, see *Langmuir* **12**, 1794–1799(1996); *J. Phys. Chem. C* **117**, 9520–9528(2013); *Inorg. Chem.* **53**, 9486–9497(2014).

Question 3: A figure including the comparison of Mo-based building blocks and their connection rules in Mo₁₃₂ ball and Mo₁₃₂ capped wheel should be given, it is beneficial for highlighting the quasi-isomers in 132-nuclearity polyoxometalates.

Answer: Thank you for your helpful suggestions. The figure including the comparison of Mo-based building blocks and their connection rules in Mo₁₃₂ ball and Mo₁₃₂

capped wheel had been given (Figure 2.2).

Figure 2.2 Polyhedral representations of Mo_{132} ball vs Mo_{132} capped wheel and their Mo-based building blocks. Color code: Mo blue/red/cyan/yellow.

Question 4: It is better for the authors to provide the experimental and simulated PXRDs, these should be clearly mentioned for crystallinity and purity of the samples.

Answer: Thank you for your helpful suggestions. The simulated and experimental PXRD of the two compounds had been provided.

Figure 2.3 The simulated and experimental PXRD patterns of **1** (left) and **2** (right).

Question 5: The authors mentioned that ‘Ce Inner and Inner-Outer binding modes had not been reported in 4f-doped gigantic Mo Wheels’, to my knowledge, Ce Inner binding modes had already been discovered in $\{Mo_{100}Ce_6\}$.

Answer: Thank you for your helpful corrections. This part has been revised as follows: “To the best of our knowledge, Ce ‘Inner-Outer’ binding modes had not been reported in 4f-doped gigantic Mo wheels”.

Question 6: In Figure 4c, check the made-up word 'inner-outer'.

Answer: Thank you for your helpful corrections. The word 'inner-outer' in Figure 4c had been revised carefully.

Figure 2.4 The 'Inner-On-Outer' binding modes of Ce centers in **2a**. The Ce and Mo centers are shown in the green and blue polyhedra, respectively.

Question 7: References: several references lack final pages and show inappropriate format, please check them.

Answer: Thank you for your helpful suggestions. The final pages and inappropriate format in references had been revised.

Question 8: To benefit a broad readership, some related works about POMs should be added in the introduction part, such as *Angew. Chem. Int. Ed.* **60**, 6382–6385(2021); *Polyoxometalates* **1**, 9140006(2022); *Polyoxometalates* **1**, 9140011(2022); *Polyoxometalates* **2**, 9140020(2023).

Answer: Thank you for your helpful suggestions. The related works about POMs had been added in the introduction part.

Thanks a lot for the referee's kindly comments!

Reviewer #3 (Remarks to the Author)

The manuscript demonstrated an unparalleled Ce-mediated molecular tailoring strategy from a closed $\{\text{Mo}_{132}\}$ capped wheel to specific product half-closed product $\{\text{Ce}_{11}\text{Mo}_{96}\}$ without altering the major framework. This is always a highly desirable strategy for customizing functional materials through precisely atomic-level tailoring and structural functionality. By this directional tailoring, the target $\{\text{Ce}_{11}\text{Mo}_{96}\}$ achieves high proton conduction, which is nearly two orders of magnitude than that of parent $\{\text{Mo}_{132}\}$. This work is interesting and of broad interest to the material and chemistry community; thorough characterization is provided for each step and mechanism. However, the following questions/comments should be addressed prior to the publication of the manuscript:

Question 1: As the new structure, PXRD compared to simulated patterns of the two compounds should be provided.

Answer: Thank you for your helpful suggestions. The simulated and experimental PXRD of the two compounds had been provided.

Figure 3.1 The simulated and experimental PXRD patterns of **1** (left) and **2** (right).

Question 2: The synthetic procedures and characterization methods should be described more detailly, such as the color changes of each step and the detailed process of proton conductivity measurements, so that the work can better be reproduced by future researchers.

Answer: Thank you for your helpful suggestions. The synthetic procedures and characterization methods had been described more detail, including the color

changes of each step and the detailed process of proton conductivity measurements as follows.

Synthetic procedures

$\text{Na}_2\text{S}_2\text{O}_4$ (0.1 g, 0.60 mmol) and $\text{CH}_3\text{COONa}\cdot 3\text{H}_2\text{O}$ (0.1 g, 0.70 mmol) were added to a solution of $\text{Na}_2\text{MoO}_4\cdot 2\text{H}_2\text{O}$ (0.176 g, 0.72 mmol) in water (12 mL). The clear mixture was then acidified to pH 2.20 with 1 M H_2SO_4 and stirred for over 30 min. Subsequently, such Mo-containing aqueous solution was transferred to a 25 Teflon-lined stainless-steel container, which was heated to 150 °C and kept for two days, then slowly cooled to room temperature. The filtrate was kept in an open 15 ml beaker for about one month. The dark black crystals were collected and washed with ethanol. Yield: 26 mg (20 % based on Mo). IR (cm^{-1}): 1620 (s), 1544 (s), 1458 (w), 1413 (s), 980 (s), 858 (m), 812 (s), 750 (w), 649 (m), 572 (s), 477 (w). Elemental analysis % calcd (found): C 2.05 (2.18), H 1.59 (1.47), S 0.69 (0.67), Mo 54.1 (53.5).

CeCl_3 (0.40 g, 1.60 mmol) and Na_2SO_4 (0.80 g, 5.64 mmol) were added under stirring to a deep green solution of **1** (0.1 g, 0.004 mmol) in H_2O (250 mL). The resulting reaction mixture was acidified to pH 2.20 with 2 M H_2SO_4 and kept at 120 °C for 3 h. After filtration of the hot solution, the deep green filtrate was stored for crystallization. After one week the dark black crystals of **2** were collected by filtration, washed with ethanol. Yield: 29 mg (25 % based on Mo). IR (cm^{-1}): 1613 (s), 1411 (s), 1127 (w), 976 (s), 873 (w), 816 (s), 724 (w), 649 (m), 558 (s). Elemental analysis % calcd (found): H 2.62 (2.56), S 1.22 (1.35), Mo 43.7 (44.8); Ce 7.32 (7.62).

Materials and characterization.

All reagents and solvents were purchased from commercial sources and used as received. IR spectra were recorded on an Alpha Centauri FTIR spectrophotometer on pressed KBr pellets in the range 400~4000 cm^{-1} . Elemental analysis of cerium and molybdenum were performed with a Leaman inductively coupled plasma (ICP) spectrometer; Carbon, sulfur and hydrogen content were determined by a Euro Vector EA3000 elemental analyzer. Water contents were determined by TG analyses on a PerkinElmer TGA7 instrument in flowing N_2 with a heating rate of 10 °C min^{-1} . A Rigaku D/max-IIB X-ray diffractometer was applied for the XRD characterizations

(scanning rate: $5^{\circ} \text{ min}^{-1}$ with $\text{Cu}_{K\alpha}$ radiation, $\lambda = 1.5418 \text{ \AA}$). UV-Vis spectra were obtained by using a 752 PC UV/Vis spectrophotometer. XPS was performed on a ESCALAB 250 spectrometer. The vacuum inside the analysis chamber was maintained at $6.2 \times 10^{-6} \text{ Pa}$ during the analysis. The morphology of crystal was characterized with SEM (FESEM; XL30, FEG, FEI Company). Raman spectra were conducted by a Jobin Yvon confocal laser Raman spectroscopy. The Mott-Schottky spots were carried out at ambient environment using the electrochemical workstation (CHI 760E) in a standard three-electrode system at frequencies of 800, 1000 and 1200 Hz. The water adsorption-desorption isotherms were measured using a Micrometrics ASAP-2020 Mautomatic specific surface area. After both crystalline samples were soaked in $\text{C}_2\text{H}_5\text{OH}$ for 2 days, the activated process was conducted with the activated temperatures of 423 K under vacuum. The single-point calculations were carried out at the (U)B3LYP/6-31G(d)(H, O)/LanI2DZ(Mo) level. All DFT calculations were carried out by the Gaussian 16 (Revision D.01) program (M. J. Frisch, G. W. Trucks, H. B. Schlegel, et al. Gaussian 16, Revision D.01, Gaussian, Inc., Wallingford CT, 2019). In situ IR spectra of both compounds under water vapor were measured on a Bruker Vertex 70 Fourier Transform Infrared Spectrometer. Each powder sample was placed at the center of an IR cell, and treated in vacuo at 423 K for 6 h. Then, powder sample was exposed to water vapor at room temperature with equilibrium pressures of 0.50 to 2.50 kPa. The equilibrium of the sorption was confirmed by the lack of changes in spectra over time. The spectra were deconvoluted with the Gaussian function.

Proton Conductivity

The dried single crystals were ground uniformly into powder. The powder samples were then put into a homemade mold with a radius of 2.50 mm. Pellets were obtained *via* compression and their thicknesses were measured using a vernier caliper. Silver glue was spread on both sides of the pellets and allowed to dry in air. The pellet sample was attached with two copper electrodes. Alternating-Current (AC) impedance measurements were performed on a multichannel electrochemical test system (1470E, Solartron) at a frequency range from 1 MHz to 0.1 Hz. The relative humidity and temperature values were set using a BPS-50CL constant temperature and humidity chamber. The bulk proton conductivity was calculated by fitting the

Nyquist plot with an electrical equivalent circuit below, which consists of a contact resistor (R1), bulk resistor (R2), capacitance (CPE1), and Warburg diffusion element (W1).

The proton conductivity was calculated using the following equation:

$$\sigma = \frac{L}{SR}$$

where σ is the proton conductivity ($S \cdot \text{cm}^{-1}$), L is the thickness of the sample (cm), S is the area of the sample (cm^2), and R is the resistance (Ω). The activation energy (E_a) was calculated according to the following equation:

$$\sigma T = \sigma_0 e^{\frac{-E_a}{k_B T}}$$

where T is the temperature (K), σ_0 is the prefactor, and k_B is the Boltzmann constant.

Question 3: The compound **2** is tailored from parent compound **1**, but its yield is low to 16 %. What happens to the other part of parent **1** during this functionalization process?

Answer: Thank you for your helpful questions. The compound **2** is tailored from parent compound **1** could be taken place in a dilute solution, the concentration of Ce^{3+} remains 8×10^{-3} mol/L and the concentration of compound **1** remains 2×10^{-5} mol/L, as the crystallization process continues, the reactants (Ce^{3+} and compound **1**) are constantly consumed and their concentrations also decrease. A much more diluted solution is not beneficial for the crystallization of compound **2**. So its yield is low to 16 %. At this level, the solution still keeps deep green (this refers the Mo Green of compound **1**) after the final crystallization of compound **2**, which we may guess that there is a certain amount of compound **1** existed in the solution. In other words, compound **1** along with cerium ions are still existed in the final crystallized solution. If the final reaction solution is centrifuged and a small amount of cerium ions are added into the centrifuged solution, compound **2** could still be obtained. If we increase the dosage of cerium ions at the beginning, compound **1** will be quickly

precipitate with Ce^{3+} cations in the solution due to the high negative charge of compound **1**, which is not conducive to the tailoring process for the formation of compound **2**. In addition, compound **2** could also be obtained when the amount of reactants was increased four-fold.

During the optimization of the reaction conditions from compound **1** to compound **2**, we found the use of additional SO_4^{2-} is necessary. When additional Na_2SO_4 was introduced into this reaction system, compound **2** could be obtained with a relatively high yield (~25%). The present yield (16%) was given without Na_2SO_4 . Comparative experiments showed that the yield keeps unchanged when Na_2SO_4 was replaced by NaCl or $\text{CH}_3\text{COONa}\cdot 3\text{H}_2\text{O}$, which we found that the ionic strength of sulfite anion in the reaction solution possesses certain influence on the synthesis. In general, the sulfite anion as inorganic ligand introduces the necessary diversity into POM systems, see *Inorg. Chem.* **53**, 9486–9497(2014); *Angew. Chem. Int. Ed.* **42**, 2085–2090(2003); *Angew. Chem. Int. Ed.* **47**, 8420–842(2008). This is in line with the structural analysis, the SO_4^{2-} are important factors during the assembly of compound **2**. Sulfite anions inside half-closed **2** provides a suitable microenvironment for trapping ‘inner’ cerium centers, which plays a key role in the formation of Ce-containing product **2**. In the future, we will still focus on how to precisely prepare polyoxometalates with high yields. Such optimization of the reaction conditions from compound **1** to compound **2** had been added into the synthesis discussion part in supporting information as follows.

“Synthesis discussion of 2:(2) the use of additional SO_4^{2-} . When additional Na_2SO_4 was introduced into this reaction system, the $\{\text{Ce}_{11}\text{Mo}_{96}\}$ product could be obtained with a relatively high yield (~25%). The yield without Na_2SO_4 is 16%. Comparative experiments showed that the yield keeps unchanged when Na_2SO_4 was replaced by NaCl or $\text{CH}_3\text{COONa}\cdot 3\text{H}_2\text{O}$, which we found that the ionic strength of sulfite anion in the reaction solution possesses certain influence on the synthesis. In general, the sulfite anion as inorganic ligand introduces the necessary diversity into POM systems, see *Inorg. Chem.* **53**, 9486–9497(2014); *Angew. Chem. Int. Ed.* **42**, 2085–2090(2003); *Angew. Chem. Int. Ed.* **47**, 8420–842(2008). This is in line with the structural analysis, the SO_4^{2-} are important factors during the assembly of $\{\text{Ce}_{11}\text{Mo}_{96}\}$. Sulfite anion inside half-closed $\{\text{Ce}_{11}\text{Mo}_{96}\}$ provides a suitable microenvironment for

trapping 'inner' cerium centers, which plays a key role in the formation of Ce-containing product $\{\text{Ce}_{11}\text{Mo}_{96}\}$."

Question 4: The TGA curve of compound **1** rises slowly during the 450 °C to 750 °C, why? What reaction might have undergone? In addition, for easier comparison, the TGA curves for **1** and **2** should be in the same range of horizontal and vertical coordinates.

Answer: Thank you for your helpful suggestions. The TGA curve of compound **1** was retested and the current TGA curve seems to be more acceptable. There is little change in the TGA curve at low temperature, which does not affect the identification of crystal water of the samples. And it seems to be acceptable from 450 °C to 750 °C now.

Figure 3.2 TGA curve for **1**. 9.38% weight loss corresponds to $\sim 122 \text{ H}_2\text{O}$.

We speculate that two possible reasons may be explained for this strange situation. (1) There is something wrong with the sample room, for example, the leakage of sample room may lead to the entrance of air, the polyoxometalate samples will react with oxygen at high temperatures that result in an increase in the TGA curve. (2) There is something wrong with reference aluminum crucible. If there are some residual impurities in reference aluminum crucible during the previous test, the loss or reaction of unknown residual impurities may affect the TGA results. Thank you for your correction, in the future, we will check the TGA curves carefully.

In addition, as your advice, for easier comparison, the TGA curves for **1** and **2** had been in the same range of horizontal and vertical coordinates, see Figure 3.2 and 3.3.

Figure 3.3 TGA curve for **2**. 14.5% weight loss corresponds to ~ 170 H₂O.

Question 5: Ce is used here as a model mediator. To be able to use other metal ions, what are the suitable characteristics of the metal ions?

Answer: Thank you for your helpful questions. *4f* Ce ions are oxophilic and display high coordination numbers as well as relatively long Ln-O bond lengths, which are usually used as strong electrophiles to replace edge-sharing or corner-sharing {Mo₂} units on the parent polyoxomolybdate framework. The incorporation of lanthanide ions not only “breaks” the symmetry of polyoxomolybdate wheel on reorganization but also adjusts its curvature and functionalities of the inner surface. For example, Ce^{III} center with the coordination configuration of a distorted monocapped square antiprism (Figure 3.4a), built from four μ_2 -O atoms and five H₂O molecules – {Ce(H₂O)₅}, joins two sets of {Mo₆} (Figure 3.4b) with four Mo–O–Ce bonds, in a manner similar to a {Mo₂} unit (both act as 4-connected rods, Figure 3.4b and 3.4c). Such coordination configuration remains the key point why lanthanide ions are easy to be incorporated into polyoxomolybdate wheels. Recent studies pointed that POM frameworks are suitable for binding metal ions with high coordination numbers, more importantly, specific coordination geometry of metal centers play a decisive role during the assembly, see *Nature* **616**, 482–487(2023). In all, we choose lanthanide ions to dissociate the Mo ball mainly due to its high coordination number (nine-coordinated) and flexible coordination environment ({Ln(H₂O)₅}, distorted

monocapped square antiprism). So we found that suitable characteristics of the metal ions used for mediator should meet the following points: relatively long M-O bond lengths, high coordination numbers and specific coordination configuration (e.g. $\{\text{Ce}(\text{H}_2\text{O})_5\}$, distorted monocapped square antiprism).

Figure 3.4 (a) Representation of the coordination geometry of 4f cations, emphasizing the distorted monocapped square antiprism coordination sphere. (b) and (c) The comparison of two adjacent $\{\text{Mo}_6\}$ units linked by $\{\text{Mo}_2\}$ and 4f centers.

Figure 3.5 IR spectra for polyoxometalate “fingerprint region” of La-, Ce- or Pr-containing samples.

Table 3.1 Some reported representative Ln-containing giant Mo-oxo clusters.

Year	Compounds	lanthanide ions	Ref.
2022	$[\text{Mo}_{64}\text{Ni}_8\text{Ln}_6\text{H}_{26}\text{O}_{200}(\text{H}_2\text{O})_{30}]^{8-}$	La, Ce, Pr	Angew. Chem. Int. Ed. e202201672 (2022)
2021	$[\text{Mo}_{124}\text{Ce}_4\text{O}_{376}(\text{H}_2\text{O})_{64}\text{H}_{12}(\text{C}_8\text{H}_{13}\text{N}_4\text{O}_3)_4]^{3-}$	Ce	Chem. Sci. 12 , 2427–2432(2021)
2020	$[\text{H}_{16}\text{Mo}_{124}\text{Ce}_4\text{O}_{376}(\text{H}_2\text{O})_{56}(\text{PMo}_{12}\text{O}_{40})(\text{C}_6\text{H}_{12}\text{N}_2\text{O}_4\text{S}_2)_4]^{5-}$	Ce	Proc. Natl. Acad. Sci 117 , 10699–10705(2020)
2019	$\{(\text{Mo}_8\text{O}_{26})\text{Mo}_{124}\text{Ce}_4\text{O}_{376}(\text{H}_2\text{O})_{60}\text{H}_{12}(\text{C}_6\text{H}_9\text{N}_3\text{O}_2)_6\}^{12-}$ $\{\text{Mo}_{124}\text{Ce}_4\text{O}_{376}(\text{H}_2\text{O})_{60}\text{H}_{12}(\text{C}_6\text{H}_{14}\text{N}_4\text{O}_2)_6\}^{8-}$	Ce	J. Am. Chem. Soc. 141 , 1242–1250(2019)
2019	$[\text{H}_{12}\text{Mo}_{128}\text{Ce}_4\text{O}_{389}(\text{H}_2\text{O})_{60}(\text{C}_5\text{H}_{13}\text{N}_2\text{O}_2)_6]^{4-}$ $[\text{H}_{14}\text{Mo}_{158.5}\text{Ce}_2\text{O}_{478}(\text{H}_2\text{O})_{81}(\text{C}_5\text{H}_{13}\text{N}_2\text{O}_2)_6]^{5-}$	Ce	Angew. Chem. Int. Ed. 58 , 10867–10872(2019)
2017	$\text{Ce}_{0.5}[\text{Ce}_6\text{H}_{16.5}\text{Mo}_{130}\text{O}_{396}(\text{H}_2\text{O})_{84}]$	Ce	Angew. Chem. Int. Ed. 56 , 9727–9731(2017)
2014	$[\text{Mo}_{100}\text{Ce}_6\text{O}_{306}\text{H}_{10}(\text{H}_2\text{O})_{71}]_2^{8-}$ $[\text{Mo}_{100}\text{Ce}_6\text{O}_{306}\text{H}_{10}(\text{H}_2\text{O})_{70}]_4^{4-}$	Ce	J. Am. Chem. Soc. 136 , 14114–14120(2014)
2010	$[\text{Mo}_{96}\text{O}_{301}(\text{H}_2\text{O})_{29}\{\text{La}(\text{H}_2\text{O})_6\}_2\{\text{La}(\text{H}_2\text{O})_5\}_6]^{22-}$	La	Inorg. Chem. 49 , 9426–9437(2010)
2006	$[\text{Mo}_{150}\text{O}_{452}\text{H}_2(\text{H}_2\text{O})_{66}\{\text{La}(\text{H}_2\text{O})_5\}_2]^{24-}$	La	J. Alloys Compd. 408 , 693–700(2006)
2002	$\{[\text{Mo}_{128}\text{Eu}_4\text{O}_{388}\text{H}_{10}(\text{H}_2\text{O})_{81}]_2\}^{20-}$	Eu	Angew. Chem. Int. Ed. 41 , 2805-2808(2002)
2000	$[\text{Mo}_{120}\text{O}_{366}(\text{H}_2\text{O})_{48}\text{H}_{12}\{\text{Pr}(\text{H}_2\text{O})_5\}_6]^{6-}$	Pr	Inorg. Chem. 39 , 3112-3113(2000)

In fact, during the preparation of $\{\text{Ce}_{11}\text{Mo}_{96}\}$, all 4f lanthanide ions (except radioelement) were introduced into the reaction system for exploring the Ln-mediated molecule tailoring process. When lanthanide ion was absent, no crystals but precursor $\{\text{Mo}_{132}\}$ was obtained. Simple 3d metal salts (e.g. Co^{2+} , Mn^{2+} , Fe^{3+} , etc.) or metal coordination complexes (e.g. $\{\text{Mn}_{12}\}$, Ni complex $[\text{Ni}_2(\mu\text{-OH}_2)(\text{O}_2\text{CCMe}_3)_4(\text{HO}_2\text{CCMe}_3)_4]$, Cr complex $[\text{Cr}_3\text{O}(\text{OOCCH}_2\text{CN})_6(\text{H}_2\text{O})_3]^+$, etc.) were also introduced to the acid solution, but no crystals was obtained. We found that

only 4f lanthanide La, Ce and Pr ions could be harvested with crystal samples. However, La- or Pr-containing samples possess rather poor crystal quality that the data are really hard to be collected although careful optimization related to the reaction conditions had been conducted. So we just report the Ce-containing samples with good crystal quality for publication. At this level, previous reports show that the Ln-containing isomers in POMs possess similar IR spectra for polyoxometalate “fingerprint region”, see *Dalton Trans.* 4423–4425(2009), *Chem. Eur. J.* **20**, 12144–12156(2014), *Chem. Eur. J.* **21**, 18168–18176(2015), etc. As show in Figure 3.5, the current La-, Ce- and Pr-containing samples possess similar IR spectra for polyoxometalate “fingerprint region” (from 700 to 1100 cm^{-1}). So we guess that Ln-mediated molecule tailoring process could be realized through La, Ce and Pr ions. Thus the lanthanide contraction results in smaller ionic radii as we cross the 4f period which becomes a key factor in the replacement of $\{\text{Mo}_2\}$ from Pr onward, see *Inorg. Chem.* **41**, 167-169(2002). This phenomenon is common in Ln-containing giant Mo-oxo clusters. Table 3.1 summarizes some reported representative Ln-containing giant Mo-oxo clusters. We found that the 4f lanthanide ions are mainly focus on early lanthanide ions or just Ce ions.

In all, the related discussions had been added into the synthesis discussion part in supporting information as follows.

“Synthesis discussion of 2:(1) the choice of Ce ions. 4f lanthanide (Ln) ions are oxophilic and display high coordination numbers as well as relatively long Ln-O bond lengths, which are usually used as strong electrophiles to replace edge-sharing or corner-sharing $\{\text{Mo}_2\}$ units on the parent polyoxomolybdate framework. For example, Ln^{III} center with the coordination configuration of a distorted monocapped square antiprism, built from four $\mu_2\text{-O}$ atoms and five H_2O molecules – $\{\text{Ln}(\text{H}_2\text{O})_5\}$, joins two sets of $\{\text{Mo}_6\}$ with four Mo–O–Ce bonds, see *J. Am. Chem. Soc.* **136**, 14114–14120(2014), in a manner similar to a $\{\text{Mo}_2\}$ unit (both act as 4-connected rods). Such coordination configuration remains the key point why lanthanide ions are easy to be incorporated into polyoxomolybdate wheels. In all, we choose lanthanide ions to dissociate the Mo ball mainly due to its high coordination number (nine) and flexible coordination environment ($\{\text{Ln}(\text{H}_2\text{O})_5\}$, distorted monocapped square antiprism). In fact, during the preparation of $\{\text{Ce}_{11}\text{Mo}_{96}\}$, all 4f lanthanide ions

(except radioelement) were introduced into the reaction system for exploring the Ln-mediated molecule tailoring process. When lanthanide ion was absent, no crystals but precursor $\{Mo_{132}\}$ was obtained. Simple *3d* metal salts (e.g. Co^{2+} , Mn^{2+} , Fe^{3+} , etc.) or metal coordination complexes (e.g. $\{Mn_{12}\}$, Ni complex $[Ni_2(\mu-OH_2)(O_2CCMe_3)_4(HO_2CCMe_3)_4]$, Cr complex $[Cr_3O(OOCCH_2CN)_6(H_2O)_3]^+$, etc.) were also introduced to the acid solution, but no crystals was obtained. We found that only *4f* lanthanide La, Ce and Pr ions could be harvested with crystal samples. However, La- or Pr-containing samples possess rather poor crystal quality that the data are really hard to be collected although careful optimization related to the reaction conditions had been conducted. So we just report the Ce-containing samples with good crystal quality for publication. At this level, previous reports show that the Ln-containing isomers in POMs possess similar IR spectra for polyoxometalate “fingerprint region”, see *Dalton Trans.* 4423–4425(2009), *Chem. Eur. J.* **20**, 12144–12156(2014), *Chem. Eur. J.* **21**, 18168–18176(2015). As show in Figure 3.5, the current La-, Ce- and Pr-containing samples possess similar IR spectra for polyoxometalate “fingerprint region”. So we guess that Ln-mediated molecule tailoring process could be realized through La, Ce and Pr ions. The lanthanide contraction results in smaller ionic radii as we cross the *4f* period which becomes a key factor in the replacement of $\{Mo_2\}$ from Pr onward. This phenomenon is common in Ln-containing giant Mo-oxo clusters, see Table 3.1. We found that the *4f* lanthanide ions are mainly focus on early lanthanide ions or just Ce ions”.

Question 6: The Figure format of the full text should be uniform, for example, the format of Figures S15 to S19 is not consistent with other Figures. In addition, the color of the Raw in XPS is so light that it is almost impossible to see if the fitted data overlaps well with them. The fitted XPS spectra of Ce in Figure S19 and Figure S37 is not matched well with the Raw data, whether this will affect the valence state determination of Ce, please check it.

Answer: Thank you for your helpful suggestions. The figure format of the full text had been revised uniformly, including the format of Figures S15 to S19. Moreover, the color of the Raw in XPS had been deepened and now it is easy to see if the fitted

data overlaps well with them. Ce XPS spectra had been reconsidered and now they are matched well with the Raw data, see Figures 3.6 and 3.7.

Figure 3.6 XPS spectrum of Ce in **2**. The peaks around 904.7 and 885.8 eV in the energy regions of Ce 3d_{3/2} and Ce 3d_{5/2} correspond to Ce³⁺.

Figure 3.7 XPS spectra of Ce in **2** before and after proton-conductive measurement.

Question 7: To improve readability, the band distance of Mo...Mo can be labelled on the Figure 1e and Figure 1g.

Answer: Thank you for your helpful suggestions. The band distance of Mo...Mo had been labeled as your advice (Figure 3.8).

Figure 3.8 The structure of unprecedented gigantic {Mo₁₃₂} capped wheel. Polyhedral and ball-and-stick representations of (a) {Mo₁₁₀} wheel, (b) the angle of capping moieties, (c) **1a**, (d) the cavity (shown grey) in **1a**, (e-g) two types of edge-sharing {Mo₂} units, (h) side view of **1a** and (i) anisotropic ligand distributions in **1a**. Color code: Mo blue/red/cyan/yellow, S orange, O grey, C pink.

Question 8: Does the two isomerism {Mo₁₃₂} have identical reduction degree (45%, Mo^V₆₀Mo^{VI}₇₂)? How did the author calculate and get this conclusion?

Answer: Thank you for your helpful questions. The calculation of reduction degree is based on the recent paper, see *Angew. Chem. Int. Ed.* e202201672(2022). The two {Mo₁₃₂} isomers indeed have identical reduction degree (45%, Mo^V₆₀Mo^{VI}₇₂). As for the classical {Mo₁₃₂} ball, see *J. Am. Chem. Soc.* **138**, 10623–10629(2016), the number of Mo^V centers is 60 according to its reported molecular formula (the electrons are located on 30 edge-sharing {Mo₂} units), so the reduction degree remains the ratio of the number of Mo^V centers to the total number of Mo^{V/VI} centers, namely 60/132 ~ 0.45 (45%). As for the {Mo₁₃₂} capped wheel, it also has 60 Mo^V centers (the electrons are delocalized not only over the ring but also the edge-sharing {Mo₂} units, see **Question 9**) based on an array of techniques as confirmed in giant wheel- and ball-shaped polyoxomolybdates, see *Chem. Soc. Rev.* **41**, 7333–7334(2012), including single crystal X-ray structure analysis, elemental analyses, redox titration, UV/Vis spectroscopy as well as bond valence sum calculations, so the reduction degree remains the ratio of the number of Mo^V centers to the total number of Mo^{V/VI} centers, namely 60/132 ~ 0.45 (45%). In all, the two {Mo₁₃₂} isomers have identical reduction degree (45%, Mo^V₆₀Mo^{VI}₇₂).

Question 9: The delocalization of the reducing electrons is the key for different optical absorptions of POMs. Authors attribute this new 'Mo Green' to delocalize reducing electrons not only over the ring but also the isolated Mo–Mo bonds in {Mo^V₂} units, how did the author get this conclusion?

Answer: Thank you for your helpful questions. Polyoxomolybdates usually possess mixed valence molybdenum (Mo^{V/VI}) centers and the reducing electrons are mainly resided in Mo^V centers, so the distribution of Mo^V centers is crucial for explaining the delocalization of electrons. The determination the number and position of reduced Mo^V centers of both compounds had already been well-presented in the part of formula determination in supporting information. To clarify this conclusion more clearly as your suggestion, we also provide suitable theoretical data to further support the delocalization of electrons. The single-point calculations were carried out at the (U)B3LYP/6-31G(d)(H, O)/LanI2DZ(Mo) level. All DFT calculations were carried out by the Gaussian 16 (Revision D.01) program (M. J. Frisch, G. W. Trucks, H. B. Schlegel, et al. Gaussian 16, Revision D.01, Gaussian, Inc., Wallingford CT, 2019). The analysis of “the delocalization of the reducing electrons” by theoretical data would support this conclusion as well as strengthen this work.

Generally, to accurately describe the electronic properties of the system, it relies on reliable methods, such as the selected functional, basis sets, and also the solvent effects. The selecting of a reliable method depends largely on the size and also the numbers of the metals in the system. For the current case, {Mo₁₃₂} capped wheel possess rather large scale and very high overall negative charge, totally more than 600 atoms are involved in both anions. To accurately simulate the electronic properties of {Mo₁₃₂} capped wheel are really a great challenge. Anyway, we have tried the single point calculation for them, but failed to self-consistent force (SCF) converge even at very low computational level. Even so, we still thought that necessary theoretical data related to the delocalization of the reducing electrons is indispensable. Instead, we focus on theoretical analysis based on typical units (part of the cluster) in {Mo₁₃₂} capped wheel that involves the reducing electrons as confirmed through bond valence sum (BVS) calculations and redox titration according to previous work, it may also give rise to helpful information to support the statement “the delocalization of the reducing electrons”. Such reasonable

simplified model for theoretical analysis seems acceptable according to some other high level POM papers, see *Energy Environ. Sci.* **9**, 1012–1023(2016), $\{P_2W_{12}\}$ was extracted as a calculation model instead of $\{P_8W_{48}\}$; *J. Am. Chem. Soc.* **131**, 16051–16053(2009), $\{Ru_4\}$ was extracted as a calculation model instead of $\{Ru_4Si_2W_{20}\}$; *Nat Commun.* **10**, 370(2019), $\{Mo_2O_2S_2\}$ was extracted as a calculation model instead of Mo-S POMs, etc. According to this kind of thoughts, $\{Mo_{132}\}$ reported here remains a capped wheel configuration, which consists of a Mo wheel and two Mo caps. The Mo^V centers in both wheel and caps should be fully considered during the analysis of the delocalization of electrons. Thus, we choose both $\{H_{11}Mo_5O_{20}\}$ unit (Figure 3.9a) in wheel and $\{H_{62}Mo_{26}O_{107}\}$ unit (Figure 3.10a) that contains reduced $\{Mo^V_2\}$ units for theoretical analysis, respectively.

Detailed theoretical analysis had been presented as follows:

(a) DFT calculation of $\{H_{11}Mo_5O_{20}\}$ unit. Previous reports show that most of Mo wheels are blue colored as the electrons (Mo^V centers) from reduction are delocalized over the wheel structure, in the central belt spanning the ring, namely $\{Mo_5O_6\}$ -type double cubanes, see *Acc. Chem. Res.* **33**, 2-10(2000), *J. Am. Chem. Soc.* **142**, 13982–13988(2020). Based on this, we selected the $\{H_{11}Mo_5O_{20}\}$ unit (Figure 3.9a) in wheel of $\{Mo_{132}\}$ capped wheel as an ideal model for theoretical analysis, in which the unsaturated oxygen is terminated with H. The single-point calculations were carried out at the (U)B3LYP/6-31G(d)(H, O)/Lanl2DZ(Mo) level. As shown in Figure 3.9b, spin density of $\{H_{11}Mo_5O_{20}\}$ shows two single electrons distribute on five Mo atoms. Also, two single occupied molecular orbitals (SOMOs) of $\{H_{11}Mo_5O_{20}\}$ are delocalized at five Mo atoms (Figure 3.9c), consistent with the above-discussion. Thus, two reduced electrons (two Mo^V centers) are delocalized over each $\{Mo_5O_6\}$ -type $\{H_{11}Mo_5O_{20}\}$ unit and all $\{Mo_5O_6\}$ -type $\{H_{11}Mo_5O_{20}\}$ units are uniformly distributed on the ring, so we found that the reducing electrons are delocalized over the ring.

Figure 3.9 (a) The calculation model {H₁₁Mo₅O₂₀} unit taken from {Mo₁₃₂} capped wheel. (b) Spin density and (c) SOMOs of {H₁₁Mo₅O₂₀}. The orbital energy is presented in parenthesis.

(b) DFT calculation of {H₆₂Mo₂₆O₁₀₇} unit. {Mo^V₂} building block localizes the two reducing electrons to form the Mo^V-Mo^V bond and creates a set of structures commonly known as the Mo browns, due to their brown colour, as seen in the well-studied {Mo₁₃₂} ball, see *Angew. Chem. Int. Ed.* **42**, 2085–2090(2003), *J. Am. Chem. Soc.* **138**, 10623–10629(2016). Based on this basis, we selected the {H₆₂Mo₂₆O₁₀₇} unit (Figure 3.10a) that contains reduced {Mo^V₂} units in {Mo₁₃₂} capped wheel as an ideal model for theoretical analysis, in which the unsaturated oxygen is terminated with H. The single-point calculations were carried out at the (U)B3LYP/6-31G(d)(H, O)/Lanl2DZ(Mo) level. The occupied molecular orbitals are located at {Mo^V₂} units, indicating obvious Mo^V...Mo^V interactions (Figure 3.10c). Furthermore, the electron density map clearly shows that the electron density distributes on all {Mo^V₂} units (Figure 3.10b). Thus, we find that reducing electrons are also localized in reduced and isolated {Mo^V₂} units of {Mo₁₃₂} capped wheel.

Figure 3.10 (a) The calculation model $\{H_{62}Mo_{26}O_{107}\}$ unit taken from $\{Mo_{132}\}$ capped wheel. (b) The electron density map of $\{H_{62}Mo_{26}O_{107}\}$ unit. (c) Occupied molecular orbitals of $\{H_{62}Mo_{26}O_{107}\}$ unit. The orbital energy is presented in parenthesis.

Based on current theoretical analysis for both rational $\{H_{11}Mo_5O_{20}\}$ units in wheel and $\{H_{62}Mo_{26}O_{107}\}$ units that contains reduced $\{Mo^{V_2}\}$ units, it is not difficult to find that the reducing electrons are not only localized in reduced and isolated $\{Mo^{V_2}\}$ units as well as delocalized over all $\{Mo_5O_6\}$ -type double cubanes in the ring. In summary, to a certain degree, if the reducing electrons are simply delocalized over the ring, it exhibits classical molybdenum blue, while if the reducing electrons are localized in reduced $\{Mo^{V_2}\}$ units at different positions, the molybdenum cluster exhibits molybdenum red or molybdenum brown, see *Angew. Chem. Int. Ed.* **61**, e202201672(2022). Thus it can be seen that when electrons are both delocalized over the wheel and reduced $\{Mo^{V_2}\}$ units, this leads to the unprecedented new 'Mo

Green' reported in current work. Thank you for your helpful questions. Related discussions had been added into the main text and supporting information.

Question 10: Please provide the electrical equivalent circuit that used for fitting the Nyquist plot and elaborate on the calculation process of resistance.

Answer: Thank you for your helpful suggestions. The electrical equivalent circuit that used for fitting the Nyquist plot had been given supporting information (see Question 2). The bulk proton conductivity was calculated by fitting the Nyquist plot with the electrical equivalent circuit, which consists of a contact resistor (R1), bulk resistor (R2), capacitance (CPE1), and Warburg diffusion element (W1), see *Chem. Eur. J.* **27**, 12137-12143(2021), *Sci. China Chem.* **64**, 959–963(2021), *Inorg. Chem. Front.* **5**, 1213-1217(2018), *Nat. Chem.* **1**, 705–710(2009), etc. The resistance value R was obtained by equivalent circuit fitting using Zview software. The resistance could be utilized for calculating the conductivity, the conductivity (σ) was calculated as $\sigma = (1/R) (L/S)$, where L is the thickness, S is the cross-sectional area (0.196 cm²) of the sample pellet and R is the value of resistance. For example, as for proton conductivity of Ce₁₁Mo₉₆ at 30°C and 98% RH, the results for fitting indicate that R2 = 32.33 Ω , thus the conductivity (σ) was calculated as 2.21×10^{-2} S cm⁻¹. Thank you for your helpful suggestions. This part had been added into supporting information.

Thanks a lot for the referee's kindly comments!

Reviewer #4 (Remarks to the Author)

The discovery of the Mo₁₃₂ capped wheel (**1**), which is “quasi-isomeric” with the previously reported Mo₁₃₂ ball, is noteworthy in the research area of giant Mo-oxo clusters. In particular, **1** contains Mo₂ reactive sites, which can be eliminated by Ce coordination to produce Ce₁₁Mo₉₆ (**2**) with an open cavity. The open cavity of Ce₁₁Mo₉₆ (**2**) leads to high water adsorption property and proton conductivity, which are comparable to those of previously reported POM-based or MOF-based proton conductors. These findings would potentially interest the readers of *Nature Communications*. However, the crystal structure of the bulk powder of **1** and **2**, which is crucial for assessing the optical properties and proton conductivities, is missing. Additionally, most of the discussions on the optical properties and proton conductivities are based on the crystal structure, without any supporting spectroscopic measurements or theoretical calculations. Therefore, major revision is required to meet the standards of publication in *Nature Communications*.

Question 1: Please provide the CHNS analysis of compounds **1** and **2**.

Answer: Thank you for your helpful suggestions. The C, H and S analysis of compounds **1** and **2** had been provided as follow. “Compound **1**, elemental analysis % calcd (found): C 2.05 (2.18), H 1.59 (1.47), S 0.69 (0.67). Compound **2**, elemental analysis % calcd (found): H 2.62 (2.56), S 1.22 (1.35).

Question 2: Please provide more details on the choice of synthetic conditions, such as the pH and temperature. Why are the yields of **1** and **2** rather low (16% and 20% based on Mo)? If you could only obtain 26 and 5 mg of **1** and **2**, respectively, from one batch, it would be quite difficult to carry out characterization and function evaluation.

Answer: Thank you for your helpful suggestions and questions.

(1) Please provide more details on the choice of synthetic conditions, such as the pH and temperature.

Answer: Thank you for your helpful suggestions. The details on the choice of synthetic conditions had been given for capped wheel {Mo₁₃₂} (including pH and

temperature) and $\{\text{Ce}_{11}\text{Mo}_{96}\}$ (mainly focus on the choice of Ce ions and the use of additional SO_4^{2-}). The detailed synthesis discussion had been shown as follows:

The capped wheel $\{\text{Mo}_{132}\}$ was synthesized from a one-pot reaction of an acidified aqueous mixture of $\text{Na}_2\text{MoO}_4 \cdot 2\text{H}_2\text{O}$ (0.72 mmol), $\text{Na}_2\text{S}_2\text{O}_4$ (0.60 mmol) and $\text{CH}_3\text{COONa} \cdot 3\text{H}_2\text{O}$ (0.70 mmol) heated in a 25ml Teflon-lined stainless-steel container at 150 °C. The combined reagents changed from a colorless mixture to transparent, blue solution before heating. The key variables for the formation of capped wheel $\{\text{Mo}_{132}\}$ were found to be both accurate pH tuning and constant high temperature. As for the pH value, high-nuclearity wheel-shaped molybdenum oxide clusters are commonly synthesized at a specific low pH value, see *Chem. Soc. Rev.* **41**, 7431-7463(2012). Thus, we analyze the formation of capped wheel $\{\text{Mo}_{132}\}$ with the pH value as the single variable: we found that increasing the pH value resulted in the oxidation of the Mo^{V} centers (white precipitate was formed), whilst much lower pH (pH < 1.0) yielded only amorphous precipitate. pH 1.8-2.5 is suitable for the assembly of capped wheel $\{\text{Mo}_{132}\}$ with its highest yield located at pH 2.2. As for the high temperature, previously reported high-nuclearity wheel-shaped molybdenum oxide clusters had been synthesized from reactions heated to between 20~90 °C. By heating the reactant to 150 °C we bring the system from standard reaction conditions to hydrothermal reaction conditions. During the synthesis of high-nuclearity polyoxometalate clusters, energy needs to be provided into the system for promoting the conversion of reactants to products, and differences in energy can lead to the generation of different structures, see *Chem. Soc. Rev.* **36**, 105–121(2007). At this stage, hydrothermal method was conducted in a high-temperature and high-pressure state, see *J. Am. Chem. Soc.* **142**, 13982–13988(2020), which is good for the construction of new molybdenum oxide clusters featuring novel building blocks or special configuration (*e.g.* capped wheel molybdenum oxide cluster). The high temperatures between 140 and 155 °C are suitable for the formation of capped wheel $\{\text{Mo}_{132}\}$ with its highest yield located at 150 °C. The results showed that hydrothermal reactions could open a whole new world to the chemistry of giant polyoxomolybdates, which had been almost exclusively prepared under ambient conditions.

The tailoring product {Ce₁₁Mo₉₆} was obtained through the reaction of an aqueous solution of capped wheel {Mo₁₃₂} with cerium chloride. During the preparation of {Ce₁₁Mo₉₆}, two important factors from {Mo₁₃₂} to {Ce₁₁Mo₉₆} during the synthesis should be emphasized: (1) the choice of Ce ions. Detailed discussions had been shown in **Question 3**. (2) the use of additional SO₄²⁻. When additional Na₂SO₄ was introduced into this reaction system, the {Ce₁₁Mo₉₆} product could be obtained with a relatively high yield (~25%). The present yield (16%) was given without Na₂SO₄. Comparative experiments showed that the yield keeps unchanged when Na₂SO₄ was replaced by NaCl or CH₃COONa·3H₂O, which we found that the ionic strength of sulfite anion in the reaction solution possesses certain influence on the synthesis. In general, the sulfite anion as inorganic ligand introduces the necessary diversity into POM systems, see *Inorg. Chem.* **53**, 9486–9497(2014); *Angew. Chem. Int. Ed.* **42**, 2085–2090(2003); *Angew. Chem. Int. Ed.* **47**, 8420–842(2008). This is in line with the structural analysis, the SO₄²⁻ are important factors during the assembly of {Ce₁₁Mo₉₆}. Sulfite anions inside half-closed {Ce₁₁Mo₉₆} provides a suitable microenvironment for trapping ‘inner’ cerium centers, which plays a key role in the formation of Ce-containing product {Ce₁₁Mo₉₆}.

(2) Why are the yields of **1** and **2** rather low (16% and 20% based on Mo)? If you could only obtain 26 and 5 mg of **1** and **2**, respectively, from one batch, it would be quite difficult to carry out characterization and function evaluation.

Answer: Thank you for your helpful questions. In polyoxometalate chemistry, the yield around 5% is considered as a rather low yield, see *Angew. Chem. Int. Ed.* **53**, 10362–10366(2014); *Angew. Chem. Int. Ed.* **49**, 4117–4120(2010); *Angew. Chem. Int. Ed.* **41**, 1162–1165(2002); *Chem. Eur. J.* **19**, 11007–11015(2013), etc. Table 4.1 provides some recent reported well-known polyoxometalate clusters published in high level journals. In this way, we may find that the yields of both 16% and 20% seem to be acceptable. Compound **1** could be obtained after quadrupling the amount of reactants with the yield remains little change. Compound **2** possesses a relatively high yield (~25%) in the presence of additional Na₂SO₄ (see **Question 2-(1)** above) and it could also be obtained when the amount of reactants was increased four-fold. So we can obtain ~0.10g and ~0.03g of **1** and **2**, respectively, from one batch. By the way, we have twenty 100ml as well as more than four hundred 25ml

Teflon-lined stainless-steel containers. Thus the samples of **1** and **2** for characterization and function evaluation could be collected within half a year. Table 4.1 also provides recent polyoxometalate clusters with similar yields (or lower yields) that are used for characterization and function evaluation. In summary, the current yields and subsequent characterization and function evaluation seem reasonable in polyoxometalate chemistry. In the future, we will focus on how to improve the yields during the design and synthesis of polyoxometalate clusters.

In all, the related discussions had been added into the synthesis discussion part in supporting information based on **Question 2** and **Question 3**. The detailed synthesis discussion had been shown in the end of **Question 3**.

Table 4.1 Some recent well-known polyoxometalate clusters with similar yields.

Year	Compounds	Yield	Functionality	Ref.
2023	$\{\text{Mo}_{132}\}/\{\text{Ce}_{11}\text{Mo}_{96}\}$	20%/25% based on Mo	proton conductivity	this work
2023	$\text{Dy}_{12}\text{Nb}_{12}\text{C}\{(\text{Nb}_6\text{O}_{19})_{12}\}$	8% based on Nb_6O_{19}	metal recognitions	Angew. Chem. Int. Ed. 62 , e202217926 (2023)
2022	$[\text{Cr}_3]_4[\text{CoW}_{12}]$	~20% yield based on W	oxygen evolution reaction	J. Am. Chem. Soc. 144 , 2980–2986(2022)
2022	$\{\text{Mo}_{126}\text{W}_{30}\}$	21.9% based on W	ligand exchange	Angew. Chem. Int. Ed. 61 , e202213910 (2022)
2022	$\{\text{SiW}_9\text{O}_{34}\text{Ni}_4\}_6\text{L}_9$	9.6% based on W	CO_2 reduction	Angew. Chem. Int. Ed. 61 , e202117637(2022)
2022	$\{\text{P}_5\text{W}_{30}\}_2\text{C}\{\text{Mo}_{22}\text{Fe}_8\}$	21.3% based on Fe	proton conductivity	Angew. Chem. Int. Ed. 61 , e202200666(2022)
2020	$\{\text{Mo}_{240}\}$	31 % based on Mo	proton conductivity	J. Am. Chem. Soc. 142 , 13982–13988(2020)

Question 3: Why did the authors focus on Ce among the lanthanides to dissociate the Mo ball?

Answer: Thank you for your helpful questions. 4f lanthanide (Ln) ions are oxophilic and display high coordination numbers as well as relatively long Ln-O bond lengths, which are usually used as strong electrophiles to replace edge-sharing or corner-

sharing $\{\text{Mo}_2\}$ units on the parent polyoxomolybdate framework. The incorporation of lanthanide ions not only “breaks” the symmetry of polyoxomolybdate wheel on reorganization but also adjusts its curvature and functionalities of the inner surface. For example, Ln^{III} center with the coordination configuration of a distorted monocapped square antiprism (Figure 4.1a), built from four $\mu_2\text{-O}$ atoms and five H_2O molecules – $\{\text{Ln}(\text{H}_2\text{O})_5\}$, joins two sets of $\{\text{Mo}_6\}$ (Figure 4.1b) with four Mo–O–Ce bonds, in a manner similar to a $\{\text{Mo}_2\}$ unit (both act as 4-connected rods, Figure 4.1b and 4.1c). Such coordination configuration remains the key point why lanthanide ions are easy to be incorporated into polyoxomolybdate wheels. Recent studies pointed that POM frameworks are suitable for binding metal ions with high coordination numbers, more importantly, specific coordination geometry of metal centers play a decisive role during the assembly, see *Nature* **616**, 482–487(2023). In all, we choose lanthanide ions to dissociate the Mo ball mainly due to its high coordination number (nine-coordinated) and flexible coordination environment ($\{\text{Ln}(\text{H}_2\text{O})_5\}$, distorted monocapped square antiprism). In particular, $\{\text{Ce}(\text{H}_2\text{O})_5\}$ moieties with Ce ‘On’ binding modes lay the foundation of the Ce-mediated molecule tailoring process from $\{\text{Mo}_{132}\}$ to $\{\text{Ce}_{11}\text{Mo}_{96}\}$ in this work.

Figure 4.1 (a) Representation of the coordination geometry of $4f$ cations, emphasizing the distorted monocapped square antiprism coordination sphere. (b) and (c) The comparison of two adjacent $\{\text{Mo}_6\}$ units linked by $\{\text{Mo}_2\}$ and $4f$ centers.

In fact, during the preparation of $\{\text{Ce}_{11}\text{Mo}_{96}\}$, all $4f$ lanthanide ions (except radioelement) were introduced into the reaction system for exploring the Ln-mediated molecule tailoring process. When lanthanide ion was absent, no crystals but precursor $\{\text{Mo}_{132}\}$ was obtained. Simple $3d$ metal salts (*e.g.* Co^{2+} , Mn^{2+} , Fe^{3+} , *etc.*)

or metal coordination complexes (e.g. $\{Mn_{12}\}$, Ni complex $[Ni_2(\mu-OH_2)(O_2CCMe_3)_4(HO_2CCMe_3)_4]$, Cr complex $[Cr_3O(OOCCH_2CN)_6(H_2O)_3]^+$, etc.) were also introduced to the acid solution, but no crystals were obtained. We found that only 4f lanthanide La, Ce and Pr ions could be harvested with crystal samples. However, La- or Pr-containing samples possess rather poor crystal quality that the data are really hard to be collected although careful optimization related to the reaction conditions had been conducted. So we just report the Ce-containing samples with good crystal quality for publication.

Figure 4.2 IR spectra for polyoxometalate “fingerprint region” of La-, Ce- or Pr-containing samples.

At this level, previous reports show that the Ln-containing isomers in POMs possess similar IR spectra for polyoxometalate “fingerprint region”, see *Dalton Trans.* 4423–4425(2009), *Chem. Eur. J.* **20**, 12144–12156(2014), *Chem. Eur. J.* **21**, 18168–18176(2015), etc. As shown in Figure 4.2, the current La-, Ce- and Pr-containing samples possess similar IR spectra for polyoxometalate “fingerprint region” (from 700 to 1100 cm^{-1}). So we guess that Ln-mediated molecule tailoring process could be realized through La, Ce and Pr ions. Thus the lanthanide contraction results in smaller ionic radii as we cross the 4f period which becomes a key factor in the replacement of $\{Mo_2\}$ from Pr onward, see *Inorg. Chem.* **41**, 167-169(2002). This phenomenon is common in Ln-containing giant Mo-oxo clusters. Table 4.2 summarizes some reported representative Ln-containing giant Mo-oxo clusters. We

found that the 4f lanthanide ions are mainly focus on early lanthanide ions or just Ce ions.

Table 4.2 Some reported representative Ln-containing giant Mo-oxo clusters.

Year	Compounds	lanthanide ions	Ref.
2022	$[\text{Mo}_{64}\text{Ni}_8\text{Ln}_6\text{H}_{26}\text{O}_{200}(\text{H}_2\text{O})_{30}]^{8-}$	La, Ce, Pr	Angew. Chem. Int. Ed. e202201672 (2022)
2021	$[\text{Mo}_{124}\text{Ce}_4\text{O}_{376}(\text{H}_2\text{O})_{64}\text{H}_{12}(\text{C}_8\text{H}_{13}\text{N}_4\text{O}_3)_4]^{3-}$	Ce	Chem. Sci. 12 , 2427–2432(2021)
2020	$[\text{H}_{16}\text{Mo}_{124}\text{Ce}_4\text{O}_{376}(\text{H}_2\text{O})_{56}(\text{PMo}_{12}\text{O}_{40})(\text{C}_6\text{H}_{12}\text{N}_2\text{O}_4\text{S}_2)_4]^{5-}$	Ce	Proc. Natl. Acad. Sci 117 , 10699–10705(2020)
2019	$\{(\text{Mo}_8\text{O}_{26})\text{Mo}_{124}\text{Ce}_4\text{O}_{376}(\text{H}_2\text{O})_{60}\text{H}_{12}(\text{C}_6\text{H}_9\text{N}_3\text{O}_2)_6\}^{12-}$ $\{\text{Mo}_{124}\text{Ce}_4\text{O}_{376}(\text{H}_2\text{O})_{60}\text{H}_{12}(\text{C}_6\text{H}_{14}\text{N}_4\text{O}_2)_6\}^{8-}$	Ce	J. Am. Chem. Soc. 141 , 1242–1250(2019)
2019	$[\text{H}_{12}\text{Mo}_{128}\text{Ce}_4\text{O}_{389}(\text{H}_2\text{O})_{60}(\text{C}_5\text{H}_{13}\text{N}_2\text{O}_2)_6]^{4-}$ $[\text{H}_{14}\text{Mo}_{158.5}\text{Ce}_2\text{O}_{478}(\text{H}_2\text{O})_{81}(\text{C}_5\text{H}_{13}\text{N}_2\text{O}_2)_6]^{5-}$	Ce	Angew. Chem. Int. Ed. 58 , 10867–10872(2019)
2017	$\text{Ce}_{0.5}[\text{Ce}_6\text{H}_{16.5}\text{Mo}_{130}\text{O}_{396}(\text{H}_2\text{O})_{84}]$	Ce	Angew. Chem. Int. Ed. 56 , 9727–9731(2017)
2014	$[\text{Mo}_{100}\text{Ce}_6\text{O}_{306}\text{H}_{10}(\text{H}_2\text{O})_{71}]_2^{8-}$ $[\text{Mo}_{100}\text{Ce}_6\text{O}_{306}\text{H}_{10}(\text{H}_2\text{O})_{70}]_4^{4-}$	Ce	J. Am. Chem. Soc. 136 , 14114–14120(2014)
2010	$[\text{Mo}_{96}\text{O}_{301}(\text{H}_2\text{O})_{29}\{\text{La}(\text{H}_2\text{O})_6\}_2\{\text{La}(\text{H}_2\text{O})_5\}_6]^{22-}$	La	Inorg. Chem. 49 , 9426–9437(2010)
2006	$[\text{Mo}_{150}\text{O}_{452}\text{H}_2(\text{H}_2\text{O})_{66}\{\text{La}(\text{H}_2\text{O})_5\}_2]^{24-}$	La	J. Alloys Compd. 408 , 693–700(2006)
2002	$\{[\text{Mo}_{128}\text{Eu}_4\text{O}_{388}\text{H}_{10}(\text{H}_2\text{O})_{81}]_2\}^{20-}$	Eu	Angew. Chem. Int. Ed. 41 , 2805–2808(2002)
2000	$[\text{Mo}_{120}\text{O}_{366}(\text{H}_2\text{O})_{48}\text{H}_{12}\{\text{Pr}(\text{H}_2\text{O})_5\}_6]^{6-}$	Pr	Inorg. Chem. 39 , 3112–3113(2000)

In all, the related discussions had been added into the synthesis discussion part in supporting information based on **Question 2** and **Question 3** as follows.

Synthesis discussion of 1: The capped wheel $\{\text{Mo}_{132}\}$ was synthesized from a one-pot reaction of an acidified aqueous mixture of $\text{Na}_2\text{MoO}_4 \cdot 2\text{H}_2\text{O}$ (0.72 mmol), $\text{Na}_2\text{S}_2\text{O}_4$ (0.60 mmol) and $\text{CH}_3\text{COONa} \cdot 3\text{H}_2\text{O}$ (0.70 mmol) heated in a 25ml Teflon-lined stainless-steel container at 150 °C. The combined reagents changed from a colorless

mixture to transparent, blue solution before heating. The key variables for the formation of capped wheel $\{\text{Mo}_{132}\}$ were found to be both accurate pH tuning and constant high temperature. On one hand, high-nuclearity wheel-shaped molybdenum oxide clusters are commonly synthesized at a specific low pH value, see *Chem. Soc. Rev.* **41**, 7431-7463(2012). Thus, we analyze the formation of capped wheel $\{\text{Mo}_{132}\}$ with pH value as the single variable: we found that increasing the pH value resulted in the oxidation of the Mo^{V} centers (white precipitate was formed), whilst much lower pH ($\text{pH} < 1.0$) yielded only amorphous precipitate. pH 1.8-2.5 is suitable for the assembly of capped wheel $\{\text{Mo}_{132}\}$ with its highest yield located at pH 2.2. On the other hand, previously reported high-nuclearity wheel-shaped molybdenum oxide clusters had been synthesized from reactions heated to between 20~90 °C. By heating the reactant to 150 °C we bring the system from standard reaction conditions to hydrothermal reaction conditions. Hydrothermal method was conducted in a high-temperature and high-pressure state, see *J. Am. Chem. Soc.* **142**, 13982–13988(2020), which is good for the construction of new molybdenum oxide clusters featuring novel building blocks or special configuration (e.g. capped wheel molybdenum oxide cluster). The high temperatures between 140 and 155 °C are suitable for the formation of capped wheel $\{\text{Mo}_{132}\}$ with its highest yield located at 150 °C. The results showed that hydrothermal reactions could open a whole new world to the chemistry of giant polyoxomolybdates, which had been almost exclusively prepared under ambient conditions. Besides, capped wheel $\{\text{Mo}_{132}\}$ could be obtained after quadrupling the amount of reactants with the yield remains little change.

Synthesis discussion of 2: During the synthesis, two important factors should be emphasized: (1) the choice of Ce ions. 4f lanthanide (Ln) ions are oxophilic and display high coordination numbers as well as relatively long Ln-O bond lengths, which are usually used as strong electrophiles to replace edge-sharing or corner-sharing $\{\text{Mo}_2\}$ units on the parent polyoxomolybdate framework. For example, Ln^{III} center with the coordination configuration of a distorted monocapped square antiprism, built from four μ_2 -O atoms and five H_2O molecules – $\{\text{Ln}(\text{H}_2\text{O})_5\}$, joins two sets of $\{\text{Mo}_6\}$ with four Mo–O–Ce bonds, see *J. Am. Chem. Soc.* **136**, 14114–14120(2014), in a manner similar to a $\{\text{Mo}_2\}$ unit (both act as 4-connected

rods). Such coordination configuration remains the key point why lanthanide ions are easy to be incorporated into polyoxomolybdate wheels. In all, we choose lanthanide ions to dissociate the Mo ball mainly due to its high coordination number (nine) and flexible coordination environment ($\{\text{Ln}(\text{H}_2\text{O})_5\}$, distorted monocapped square antiprism). In fact, during the preparation of $\{\text{Ce}_{11}\text{Mo}_{96}\}$, all 4f lanthanide ions (except radioelement) were introduced into the reaction system for exploring the Ln-mediated molecule tailoring process. When lanthanide ion was absent, no crystals but precursor $\{\text{Mo}_{132}\}$ was obtained. Simple 3d metal salts (e.g. Co^{2+} , Mn^{2+} , Fe^{3+} , etc.) or metal coordination complexes (e.g. $\{\text{Mn}_{12}\}$, Ni complex $[\text{Ni}_2(\mu\text{-OH}_2)(\text{O}_2\text{CCMe}_3)_4(\text{HO}_2\text{CCMe}_3)_4]$, Cr complex $[\text{Cr}_3\text{O}(\text{OOCCH}_2\text{CN})_6(\text{H}_2\text{O})_3]^+$, etc.) were also introduced to the acid solution, but no crystals was obtained. We found that only 4f lanthanide La, Ce and Pr ions could be harvested with crystal samples. However, La- or Pr-containing samples possess rather poor crystal quality that the data are really hard to be collected although careful optimization related to the reaction conditions had been conducted. So we just report the Ce-containing samples with good crystal quality for publication. At this level, previous reports show that the Ln-containing isomers in POMs possess similar IR spectra for polyoxometalate “fingerprint region”, see *Dalton Trans.* 4423–4425(2009), *Chem. Eur. J.* **20**, 12144–12156(2014), *Chem. Eur. J.* **21**, 18168–18176(2015). As show in Figure 4.2, the current La-, Ce- and Pr-containing samples possess similar IR spectra for polyoxometalate “fingerprint region”. So we guess that Ln-mediated molecule tailoring process could be realized through La, Ce and Pr ions. The lanthanide contraction results in smaller ionic radii as we cross the 4f period which becomes a key factor in the replacement of $\{\text{Mo}_2\}$ from Pr onward. This phenomenon is common in Ln-containing giant Mo-oxo clusters, see Table 4.2. We found that the 4f lanthanide ions are mainly focus on early lanthanide ions or just Ce ions.

(2) the use of additional SO_4^{2-} . When additional Na_2SO_4 was introduced into this reaction system, the $\{\text{Ce}_{11}\text{Mo}_{96}\}$ product could be obtained with a relatively high yield (~25%). The yield without Na_2SO_4 is 16%. Comparative experiments showed that the yield keeps unchanged when Na_2SO_4 was replaced by NaCl or $\text{CH}_3\text{COONa}\cdot 3\text{H}_2\text{O}$, which we found that the ionic strength of sulfite anion in the reaction solution possesses certain influence on the synthesis. In general, the sulfite

anion as inorganic ligand introduces the necessary diversity into POM systems, see *Inorg. Chem.* **53**, 9486–9497(2014); *Angew. Chem. Int. Ed.* **42**, 2085–2090(2003); *Angew. Chem. Int. Ed.* **47**, 8420–842(2008). This is in line with the structural analysis, the SO_4^{2-} are important factors during the assembly of $\{\text{Ce}_{11}\text{Mo}_9\}$. Sulfite anion inside half-closed $\{\text{Ce}_{11}\text{Mo}_9\}$ provides a suitable microenvironment for trapping ‘inner’ cerium centers, which plays a key role in the formation of Ce-containing product $\{\text{Ce}_{11}\text{Mo}_9\}$.

Question 4: There are two 132-Mo-atoms clusters in this paper; Mo_{132} ball and Mo_{132} capped wheel. Please provide the definition of “quasi-isomers” and “quasi-isomerism”. Please also compare the synthetic conditions of the isomers.

Answer: Thank you for your helpful suggestions.

(1) Please provide the definition of “quasi-isomers” and “quasi-isomerism”.

Answer: Thank you for your helpful suggestions. In chemistry, isomers are molecules with identical formulas but distinct structures. The subject of isomerism has been well established in atomically precise nanoclusters, mainly focused on Au, Ag and Cu nanoclusters, see *J. Am. Chem. Soc.* **138**, 1482–1485(2016), *J. Am. Chem. Soc.* **140**, 42, 13590–13593(2018), *Chem. Sci.* **10**, 8685–8693(2019), etc. These Au, Ag and Cu cluster structural isomers are those clusters that follow the same compositions but adopt different configurations. In the strictest point of view, all elements including metals and ligands in these nanocluster structural isomers should be identical. However, if the isomers share the same number of metal centers with different geometrical structures and capping ligands, they could be considered as “quasi-isomers”, see *Chem. Mater.* **33**, 39–62(2021). Molecular isomerism in polyoxometalates is rather rare, see *Chem. Eur. J.* **19**, 2976–2981(2013), *Angew. Chem. Int. Ed.* **57**, 2972–2975(2018), in this regard, according to the definition of isomers and quasi-isomers in atomically precise nanoclusters, we concluded that “polyoxometalate quasi-isomers” may share same number of metal centers with similar metal-oxygen bonding modes (e.g. the same valence state of metal centers and characteristic building blocks) by ignoring different outer ligands. In this work, 132-Mo-atom clusters possess identical reduction degree (45%, $\text{Mo}^{\text{V}}_{60}\text{Mo}^{\text{VI}}_{72}$) and characteristic $\{\text{Mo}(\text{Mo})_5\}$ motifs with similar connection rules but having different

geometrical structures (*e.g.* ball vs. capped wheel), thus we think that they are 132-Mo-atom “quasi-isomers”. This pair 132-Mo-atom “quasi-isomers” constitutes quasi-isomerism in polyoxometalates. We speculate that the definition of isomerism in polyoxometalates will become more appropriate with the discovery of more polyoxometalate isomers and quasi-isomers. Thank you for your suggestion, this part has been included in the appropriate sections in main text and supporting information.

(2) Please also compare the synthetic conditions of the isomers.

Answer: Thank you for your helpful suggestions. The comparison of the synthetic conditions for such isomers may provide meaningful evidence for synthesis of other polyoxomolybdate isomers in the future. The Na-salt of {Mo₁₃₂} ball was obtained by the reaction of Na₂MoO₄·2H₂O (0.628 mmol), N₂H₆SO₄ (0.123 mmol) and CH₃COONa (3.243 mmol) in 5 ml water at room temperature, see *J. Am. Chem. Soc.* **138**, 10623–10629(2016), while the Na-salt of {Mo₁₃₂} capped wheel was synthesized from a one-pot reaction of an acidified aqueous mixture of Na₂MoO₄·2H₂O (0.72 mmol), Na₂S₂O₄ (0.60 mmol) and CH₃COONa·3H₂O (0.70 mmol) upon heating. The synthetic conditions of the isomers remain many commonalities, such as the molybdenum raw material, acidic environment, the existence of reducing agent, and the assistance of small organic ligands (*e.g.* CH₃COO⁻), *etc.* However, the differences for the synthetic conditions are much worth discussing, which had been summarized as the following four tips: (a) different reaction temperatures. {Mo₁₃₂} ball was synthesized at room temperature (without heating) and {Mo₁₃₂} capped wheel could only be obtained under hydrothermal conditions. This is in line with the fact that hydrothermal reactions may bring novel building blocks or special structure of molybdenum oxide clusters (*e.g.* capped wheel configuration). (b) different pH value. Wheel-shaped molybdenum oxide clusters are commonly synthesized at a specific low pH value (< 2.5), see *Chem. Soc. Rev.* **41**, 7431-7463(2012), and {Mo₁₃₂} ball was obtained at pH ~4.00, pH value is a known crucial parameter in polyoxomolybdate chemistry as a series of available Mo-based building units toward the formation of the final structures are driven by it. (c) different reducing agent. A series of hydrazide-containing compounds had been used in the construction of high-nuclearity molybdenum oxide clusters, including {Mo₁₃₂} ball. Sodium hydrosulfite

had been employed as effective reducing agent in the construction of high-nuclearity molybdenum oxide clusters, as our previous work, see *Chem. Sci.* **13**, 4573-4580(2022). (d) different concentrations of Mo-containing solution. The concentration of Mo-containing solution for {Mo₁₃₂} ball is a little high than that for {Mo₁₃₂} capped wheel. Higher concentrations were found to promote crystal growth quickly, while lower concentrations lead to good crystals but with a longer crystallization time and lower yields, see *J. Am. Chem. Soc.* **141**, 1242–1250(2019). An independent synthesis between {Mo₁₃₂} ball and {Mo₁₃₂} capped wheel had been added into supporting information.

Question 5: The UV-vis spectra and Moss-Schottky measurements show that the Mo₁₃₂ ball and the Mo₁₃₂ capped wheel show different optical behaviors, which the authors attribute to “the difference in delocalization of the reducing electrons”. Could you please provide experimental or theoretical data (such as electronic band structure calculations) to support this hypothesis?

Answer: Thank you for your helpful suggestions. We also approve that the analysis of “the difference in delocalization of the reducing electrons” by theoretical data would support this hypothesis as well as strengthen this work. In practical terms, the exploration of electronic distributions of {Mo₁₃₂} ball and the {Mo₁₃₂} capped wheel may give rise to favorable evidence for its delocalization of the reducing electrons. In fact, theoretical calculation based on density functional theory (DFT) has become applicable to such complicated POMs-based molecular systems. The electronic distribution properties for POMs has been fairly well understood through the analysis of their electron density map, by the way, most POMs are relative simple molecules with heavy metals less than 20, see *nature* **515**, 545–549(2014); *Angew. Chem. Int. Ed.* **61**, e202112915(2022), *Nat Commun.* **11**, 490(2020), *J. Am. Chem. Soc.* **124**, 12574-12582(2002), *etc.* Thus, the theoretical calculation remains the most suitable technique for the exploration of the electronic distributions of {Mo₁₃₂} ball and {Mo₁₃₂} capped wheel. Generally, to accurately describe the electronic properties of the system, it relies on reliable methods, such as the selected functional, basis sets, and also the solvent effects. The selecting of a reliable method depends largely on the size and also the numbers of the metals in the system. For

the current case, {Mo₁₃₂} ball and {Mo₁₃₂} capped wheel possess rather large scale and very high overall negative charge, totally more than 600 atoms are involved in both anions. To accurately simulate the electronic properties (or electronic band structure calculations as your advice) of {Mo₁₃₂} ball and the {Mo₁₃₂} capped wheel are really a great challenge. Anyway, we have tried the single point calculation for them, but failed to self-consistent force (SCF) converge even at very low computational level.

Even so, we still thought that necessary theoretical data related to the delocalization of the reducing electrons is indispensable. Instead, we focus on theoretical analysis based on typical units (part of the cluster) in {Mo₁₃₂} ball and the {Mo₁₃₂} capped wheel that involves the reducing electrons as confirmed through bond valence sum (BVS) calculations and redox titration according to previous work, it may also give rise to helpful information to support the statement “the difference in delocalization of the reducing electrons”. Such reasonable simplified model for theoretical analysis seems acceptable according to some other high level POM papers, see *Energy Environ. Sci.* **9**, 1012–1023(2016), {P₂W₁₂} was extracted as a calculation model instead of {P₈W₄₈}; *J. Am. Chem. Soc.* **131**, 16051–16053(2009), {Ru₄} was extracted as a calculation model instead of {Ru₄Si₂W₂₀}; *Nat Commun.* **10**, 370(2019), {Mo₂O₂S₂} was extracted as a calculation model instead of Mo-S POMs, etc. According to this kind of thoughts, we choose {H₄₂Mo₁₆O₆₇} units (Figure 4.3a) in {Mo₁₃₂} ball for theoretical analysis, and in {Mo₁₃₂} capped wheel, we choose both {H₁₁Mo₅O₂₀} units (Figure 4.4a) in wheel and {H₆₂Mo₂₆O₁₀₇} units (Figure 4.5a) that contains reduced {Mo^V₂} units for theoretical analysis (the reasons are shown below), respectively.

Detailed theoretical analysis had been presented as follows:

(a) {Mo₁₃₂} ball. This structure possesses a metal-oxo framework with high crystal symmetry that contains 132 Mo atoms consisting of twelve {Mo₆} pentagonal units and thirty {Mo₂} linkers. As previous report, the reducing electrons in {Mo₁₃₂} ball are localized in reduced and isolated {Mo^V₂} units, see *Bull. Jpn. Soc. Coord. Chem.* **78**, 11-17(2021), *Angew. Chem. Int. Ed.* **42**, 2085–2090(2003). So we selected the classical {H₄₂Mo₁₆O₆₇} unit (Figure 4.3a) involving five {Mo^V₂} and one central {(Mo)Mo₅} unit in {Mo₁₃₂} ball as reasonable models for theoretical analysis, in which

the unsaturated oxygen is terminated with H. The single-point calculations were carried out at the (U)B3LYP/6-31G(d)(H, O)/LanI2DZ(Mo) level. Figure 4.3c shows the occupied molecular orbitals of the classical $\{H_{42}Mo_{16}O_{67}\}$ unit in $\{Mo_{132}\}$ ball, obvious $Mo^V \cdots Mo^V$ interactions could be identified. The electron density map of $\{H_{42}Mo_{16}O_{67}\}$ unit in $\{Mo_{132}\}$ ball (Figure 4.3b) highlights that the distribution of electron density is mainly focus on $\{Mo^V_2\}$ units as expected. Based on current theoretical analysis, it is not difficult to find that the reducing electrons are localized in reduced and isolated $\{Mo^V_2\}$ units in $\{Mo_{132}\}$ ball.

Figure 4.3 (a) The calculation model $\{H_{42}Mo_{16}O_{67}\}$ unit taken from $\{Mo_{132}\}$ ball. (b) The electron density map of $\{H_{42}Mo_{16}O_{67}\}$ unit. (c) Occupied molecular orbitals of $\{H_{42}Mo_{16}O_{67}\}$ unit. The orbital energy is presented in parenthesis.

(b) $\{Mo_{132}\}$ capped wheel. This structure reported here possesses a capped wheel configuration, which consists of a Mo wheel and two Mo caps. The Mo^V centers in both wheel and caps should be fully considered during the analysis of the delocalization of electrons. In fact, the number and position of reduced Mo^V centers of both compounds had already been well-presented in the part of formula determination in supporting information. Based on this, we selected the $\{H_{11}Mo_5O_{20}\}$

unit (Figure 4.4a) in wheel and $\{H_{62}Mo_{26}O_{107}\}$ (Figure 4.5a) that contains reduced $\{Mo^V_2\}$ units as ideal models for theoretical analysis, in which the unsaturated oxygen is terminated with H. The single-point calculations were carried out at the (U)B3LYP/6-31G(d)(H, O)/Lanl2DZ(Mo) level. As for the $\{H_{11}Mo_5O_{20}\}$ unit in Mo wheel part, previous reports show that most of Mo wheel are blue colored as the electrons from reduction are delocalized over the wheel structure, in the central belt spanning the ring, namely $\{Mo_5O_6\}$ -type double cubanes, which is confirmed by bond valence sum calculations, redox titration as well as UV/Vis spectroscopy, see *Acc. Chem. Res.* **33**, 2-10(2000), *J. Am. Chem. Soc.* **142**, 13982–13988(2020). As shown in Figure 4.4b, spin density of $\{H_{11}Mo_5O_{20}\}$ shows two single electrons distribute on five Mo atoms. Also, two single occupied molecular orbitals (SOMOs) of $\{H_{11}Mo_5O_{20}\}$ are delocalized at five Mo atoms (Figure 4.4c), consistent with the above-discussion. As for the $\{H_{62}Mo_{26}O_{107}\}$ unit (Figure 4.5a), which contains ten sets of $\{Mo^V_2\}$ units, the reduced electrons are expected to be localized in $\{Mo^V_2\}$ units. The occupied molecular orbitals are located at $\{Mo^V_2\}$ units, indicating obvious $Mo^V \cdots Mo^V$ interactions (Figure 4.5c). Furthermore, the electron density map clearly shows that the electron density distributes on all $\{Mo^V_2\}$ units (Figure 4.5b). Based on current theoretical analysis for both rational wheel and cap species in $\{Mo_{132}\}$ capped wheel, it is not difficult to find that the reducing electrons are not only localized in reduced and isolated $\{Mo^V_2\}$ units as well as delocalized over all $\{Mo_5O_6\}$ -type double cubanes in the ring.

Figure 4.4 (a) The calculation model $\{H_{11}Mo_5O_{20}\}$ unit taken from $\{Mo_{132}\}$ capped wheel. (b) Spin density and (c) SOMOs of $\{H_{11}Mo_5O_{20}\}$. The orbital energy is presented in parenthesis.

Figure 4.5 (a) The calculation model $\{H_{62}Mo_{26}O_{107}\}$ unit taken from $\{Mo_{132}\}$ capped wheel. (b) The electron density map of $\{H_{62}Mo_{26}O_{107}\}$ unit. (c) Occupied molecular orbitals of $\{H_{62}Mo_{26}O_{107}\}$ unit. The orbital energy is presented in parenthesis.

In summary, to a certain degree, the current theoretical data support the fact that Mo_{132} ball and the Mo_{132} capped wheel show different delocalization of the reducing electrons. If the reducing electrons are simply delocalized over the ring, it exhibits classical molybdenum blue, while if the reducing electrons are localized in reduced $\{Mo^V_2\}$ units at different positions, the molybdenum cluster exhibits molybdenum red or molybdenum brown, see *Angew. Chem. Int. Ed.* **61**, e202201672(2022). Thus it can be seen that when electrons are both delocalized over the wheel and reduced $\{Mo^V_2\}$ units, this leads to the unprecedented new ‘Mo Green’ reported in current work. From this pair of 132-Mo-atom cluster, we also found that the color displayed by reduced giant polyoxomolybdates could not only

be related to the degree of Mo reduction but also the factor of the location of reduction. Thank you for your helpful suggestions. Related discussions had been added into the main text and supporting information.

This part had been added into the main text as follows: “.....As for the {Mo₁₃₂} ball (Mo Brown), the electrons are localized in reduced and isolated {Mo^V₂} units, which is further confirmed by the electron density map and molecular orbitals of {Mo^V₂}-containing {Mo₁₆} unit in {Mo₁₃₂} ball using density functional theory (DFT) calculations (Figure 4.3 and related discussions). When the reducing electrons are delocalized not only over the ring but also the isolated {Mo^V₂} units, this contributes to the capped wheel **1a** defined as a brand new ‘Mo Green’ family. The electron density maps and molecular orbitals of reasonable calculation models in **1a**, including {Mo₅} species (Figure 4.4 and related discussions) in wheel and {Mo₂₆} species (Figure 4.5 and related discussions) that contains reduced {Mo^V₂} units, highlight that the distributions of reducing electrons are localized in {Mo₅O₆} double cubane of ring and isolated {Mo^V₂} units as expected, respectively. Therefore, from this pair of 132-Mo-atom cluster, we found that the color displayed by reduced giant polyoxomolybdates could not only be related to the degree of Mo reduction but also the factor of the location of reduction.”

This part had been added into supporting information as follows: The analysis of electronic distribution properties for this pair of 132-Mo-atom POMs by theoretical data may give rise to valuable information to clarify “their difference in delocalization of the reducing electrons”. We choose {H₄₂Mo₁₆O₆₇} units (Figure 4.3a) in {Mo₁₃₂} ball for theoretical analysis, and in {Mo₁₃₂} capped wheel, we choose both {H₁₁Mo₅O₂₀} units (Figure 4.4a) in wheel and {H₆₂Mo₂₆O₁₀₇} units (Figure 4.5a) that contains reduced {Mo^V₂} units for theoretical analysis (the reasons are shown below), respectively. Detailed theoretical analysis had been presented as follows:

(a) {Mo₁₃₂} ball. This structure possesses a metal-oxo framework with high crystal symmetry that contains 132 Mo atoms consisting of twelve {Mo₆} pentagonal units and thirty {Mo₂} linkers. As previous report, the reducing electrons in {Mo₁₃₂} ball are localized in reduced and isolated {Mo^V₂} units, see *Bull. Jpn. Soc. Coord. Chem.* **78**, 11-17(2021), *Angew. Chem. Int. Ed.* **42**, 2085–2090(2003). So we selected the classical {H₄₂Mo₁₆O₆₇} unit (Figure 4.3a) involving five {Mo^V₂} and one central

{(Mo)Mo₅} unit in {Mo₁₃₂} ball as reasonable models for theoretical analysis, in which the unsaturated oxygen is terminated with H. The single-point calculations were carried out at the (U)B3LYP/6-31G(d)(H, O)/LanI2DZ(Mo) level. Figure 4.3c shows the occupied molecular orbitals of the classical {H₄₂Mo₁₆O₆₇} unit in {Mo₁₃₂} ball, obvious Mo^V...Mo^V interactions could be identified. The electron density map of {H₄₂Mo₁₆O₆₇} unit in {Mo₁₃₂} ball (Figure 4.3b) highlights that the distribution of electron density is mainly focus on {Mo^V₂} units as expected. Based on current theoretical analysis, it is not difficult to find that the reducing electrons are localized in reduced and isolated {Mo^V₂} units in {Mo₁₃₂} ball.

(b) {Mo₁₃₂} capped wheel. This structure reported here possesses a capped wheel configuration, which consists of a Mo wheel and two Mo caps. The Mo^V centers in both wheel and caps should be fully considered during the analysis of the delocalization of electrons. In fact, the number and position of reduced Mo^V centers of both compounds had already been well-presented in the part of formula determination. Based on this, we selected the {H₁₁Mo₅O₂₀} unit (Figure 4.4a) in wheel and {H₆₂Mo₂₆O₁₀₇} (Figure 4.5a) that contains reduced {Mo^V₂} units as ideal models for theoretical analysis, in which the unsaturated oxygen is terminated with H. The single-point calculations were carried out at the (U)B3LYP/6-31G(d)(H, O)/LanI2DZ(Mo) level. As for the {H₁₁Mo₅O₂₀} unit in Mo wheel part, previous reports show that most of Mo wheel are blue colored as the electrons from reduction are delocalized over the wheel structure, in the central belt spanning the ring, namely {Mo₅O₆}-type double cubanes, which is confirmed by bond valence sum calculations, redox titration as well as UV/Vis spectroscopy, see *Acc. Chem. Res.* **33**, 2-10(2000), *J. Am. Chem. Soc.* **142**, 13982–13988(2000). As shown in Figure 4.4b, spin density of {H₁₁Mo₅O₂₀} shows two single electrons distribute on five Mo atoms. Also, two single occupied molecular orbitals (SOMOs) of {H₁₁Mo₅O₂₀} are delocalized at five Mo atoms (Figure 4.4c), consistent with the above-discussion. As for the {H₆₂Mo₂₆O₁₀₇} unit (Figure 4.5a), which contains ten sets of {Mo^V₂} units, the reduced electrons are expected to be localized in {Mo^V₂} units. The occupied molecular orbitals are located at {Mo^V₂} units, indicating obvious Mo^V...Mo^V interactions (Figure 4.5c). Furthermore, the electron density map clearly shows that the electron density distributes on all {Mo^V₂} units (Figure 4.5b). Based on current theoretical analysis for both rational

wheel and cap species in $\{\text{Mo}_{132}\}$ capped wheel, it is not difficult to find that the reducing electrons are not only localized in reduced and isolated $\{\text{Mo}^{\text{V}}_2\}$ units as well as delocalized over all $\{\text{Mo}_5\text{O}_6\}$ -type double cubanes in the ring.

In summary, to a certain degree, the current theoretical data support the fact that Mo_{132} ball and the Mo_{132} capped wheel show different delocalization of the reducing electrons. If the reducing electrons are simply delocalized over the ring, it exhibits classical molybdenum blue, while if the reducing electrons are localized in reduced $\{\text{Mo}^{\text{V}}_2\}$ units at different positions, the molybdenum cluster exhibits molybdenum red or molybdenum brown, see *Angew. Chem. Int. Ed.* **61**, e202201672(2022). Thus it can be seen that when electrons are both delocalized over the wheel and reduced $\{\text{Mo}^{\text{V}}_2\}$ units, this leads to the unprecedented new 'Mo Green' reported in current work. From this pair of 132-Mo-atom cluster, we also found that the color displayed by reduced giant polyoxomolybdates could not only be related to the degree of Mo reduction but also the factor of the location of reduction.

Question 6: Bulk powder PXRD data for **1** and **2** should be shown and compared with simulated patterns obtained from single crystal data.

Answer: Thank you for your helpful suggestions. Bulk powder PXRD data for **1** and **2** had been obtained from single crystal data. See **Question 7**.

Question 7: The structural stability after the proton conduction measurements is evaluated by IR and Raman spectra. However, these data can only provide information on the molecular structure. Therefore, PXRD data after the proton conductivity measurement are necessary to assess the structural stability of the compounds under the measurement conditions.

Answer: Thank you for your helpful suggestions. PXRD data after the proton conductivity measurement had been given as follows.

Figure 4.6 Powder X-ray patterns (PXRD) of **1**, calculated (black), crystalline sample (red), and after the proton-conductive measurement (blue).

Figure 4.7 Powder X-ray patterns (PXRD) of **2**, calculated (black), crystalline sample (red), and after the proton-conductive measurement (blue).

Question 8: Please show the equivalent circuit for fitting the Nyquist plot.

Answer: Thank you for your helpful suggestions. The electrical equivalent circuit that used for fitting the Nyquist plot had been given below, which consists of a contact resistor (R_1), bulk resistor (R_2), capacitance (CPE_1), and Warburg diffusion element (W_1). This part had been added into supporting information.

Figure 4.8 The equivalent circuit model used for fitting the Nyquist plots.

Question 9: Readers may find it interesting if the “quasi-isomer effect” of the 132-Mo-atoms clusters were presented.

Answer: Thank you for your helpful suggestions. We also think that it is interesting to show the “quasi-isomer effect” of the 132-Mo-atoms clusters. In this work, the discussions of the “quasi-isomer effect” of the 132-Mo-atoms clusters may focus on their proton conductivities and optical behaviors.

As for their proton conductivities, the proton conductivities of {Mo₁₃₂} ball and {Mo₁₃₂} capped wheel had been both tested under relative humidity (RH, 98%) and 60°C temperature. The conductivity values of {Mo₁₃₂} ball and {Mo₁₃₂} capped wheel at 60°C (98% RH) were measured to be $7.91 \times 10^{-4} \text{ S cm}^{-1}$ and $8.30 \times 10^{-4} \text{ S cm}^{-1}$, respectively. Although {Mo₁₃₂} ball has an open hollow cavity (the hollow cavity of {Mo₁₃₂} capped wheel is sealed) which is favourable for proton transfer, the chemical environments of inter-molecular in {Mo₁₃₂} capped wheel show that each pair of adjacent {Mo₁₃₂} shares a much shorter O···O distance in 3D continuous array compared to that in {Mo₁₃₂} ball, such frameworks featuring condensed packing modes are beneficial for effective proton hopping and transfer. In this regard, they possess similar proton conductivity values and little valuable “quasi-isomer effect” of the 132-Mo-atoms clusters could be provided.

As for their optical behaviors, the UV-vis spectra and Moss-Schottky measurements show that the {Mo₁₃₂} ball and the {Mo₁₃₂} capped wheel show different optical behaviors, as presented in **Question 5**, this part had been discussed tried our best based on theoretical and experimental evidences as your advice. The different optical behaviors may originate from the “quasi-isomer effect” of the 132-Mo-atoms clusters. This pair of ‘molybdenum framework isomer’ exhibits disparate optical behaviors, which may act as cluster models that bridge the gap between molecular isomerism and phase isomerism of molybdenum oxide for the purpose of better understanding both structures and photochemical performance of phase-

dependent nanomaterials. So we just provide this part in the submitted manuscript. Thank you for your helpful suggestions, in the future, we will still pay close attention to the “quasi-isomer effect” of the reported 132-Mo-atoms clusters in other application fields.

Question 10: The authors’ statement that “the dense and extended hydrogen-bonding network of **2** promotes proton transport” is inconsistent with the higher activation energy for proton conduction in **2** than in **1**.

Answer: Thank you for your helpful questions. The activation energy for proton conduction in **2** (red line) is indeed lower than the activation energy for proton conduction in **1** (blue line) as shown in Figure 5e in the main text. The statement that “the dense and extended hydrogen-bonding network of **2** promotes proton transport” is consistent with the higher activation energy for proton conduction in **1** than in **2**.

Question 11: Related with (10), the proton conduction mechanism should be supported by spectroscopic evidence, rather than solely based on the crystal structure. Please consider in-situ IR measurements under water vapor (*ACS Applied Mater Interfaces* 2021, 13, 19138), DQF 1H NMR (*JACS* 2016, 138, 27, 8505; *JACS* 2021 143, 21433), or neutron scattering techniques (*J. Phys. Chem. Lett.* 2018, 9, 5772).

Answer: Thank you for your helpful suggestions. According to your advice, we chose in situ IR measurements under water vapor as spectroscopic evidence for supporting the proton conduction mechanism. The states of water have been studied by spectroscopic methods such as IR, it has been reported that the positions of the $\nu(\text{OH})$ bands distinguish the hydrogen-bonds and characterize the states of water, see *Angew. Chem. Int. Ed.* **41**, 48(2002), *Science* **265**, 75(1994), *Science* **304**, 1134(2004), etc. In POM area, Uchida *et al.* have done pioneered research about in-situ IR measurements under water vapor conditions, see *ACS Appl. Mater. Interfaces* **13**, 19138–19147(2021), *Phys. Chem. Chem. Phys.* **19**, 29077–29083(2017), *J. Phys. Chem. C* **111**, 8218–8227(2007). Recently, Uchida *et al.* also reported some other high level POM papers, see *J. Am. Chem. Soc.* **144**, 2980–2986(2022), *Angew. Chem.*

Int. Ed. **61**, e202200666(2022), *Chem. Sci.* **14**, 5453–5459(2023), *Small* **13**, 2300743(2023), *Coord. Chem. Rev.* **462**, 214524(2022), *etc.* We learnt a lot from these papers and all of them had been cited in the revised main text. In this work, in situ IR spectra of both compounds under water vapor were measured on a Bruker Vertex 70 Fourier Transform Infrared Spectrometer. Figure 4.9 displays the background-subtracted IR spectra of compounds **1** and **2** under a water vapor pressure of 2.50 kPa ($P/P_0 = 0.90$). The absorption bands at 3000–3700 and 1600 cm^{-1} can be assigned to water molecules with OH stretching ($\nu(\text{OH})$) and HOH bending ($\delta(\text{HOH})$), respectively, see *ACS Appl. Mater. Interfaces* **13**, 19138–19147(2021), *J. Phys. Chem. C* **111**, 8218–8227(2007). It has been demonstrated that the band with regard to $\delta(\text{HOH})$ of water molecules remains less responsive to hydrogen-bonding networks, see *Science* **344**, 1009–1012(2014).

Figure 4.9 In situ IR spectra of **1** (blue) and **2** (red) under a water vapor pressure of 2.50 kPa.

More importantly, previous works pointed that the $\nu(\text{OH})$ bands of water molecules in a hydrogen bonding environment appear separately and the band positions, especially the broad bands appeared in the range of 3000–3700 cm^{-1} , decrease with the increase in the number and strength of hydrogen bonds, see *J. Phys. Chem. A* **93**, 4837–4843(1989), *Science* **304**, 1134(2004), *etc.* The characteristic broad bands appeared gradually after introducing water vapor (Figure 4.10). Figure 4.11 highlights the hydroxyl stretching region of the spectra, the observed bands

could be deconvoluted into four Gaussian peaks centered at (i) 3513–3569 cm^{-1} , (ii) 3327–3403 cm^{-1} , (iii) 3155–3236 cm^{-1} , and (iv) 2988–3012, which may be classified into the following four groups based on the water molecules in crystal structures of **1-2** as well as previous study about the state of water molecules in Keggin-type and Preyssler-type POMs system reported by Uchida *et al.*: the $\nu(\text{OH})$ bands of water (a) in the vicinity of a metal ion (Na^+ or Ce^{3+}) and without hydrogen-bonds, (b) in the vicinity of a metal ion (Na^+ or Ce^{3+}) and at a hydrogen-bonding distance with an oxide ion or an oxygen atom, (c) in hydrogen-bonding distances with two oxygen atoms and/or oxide ion, and in (c) featuring short hydrogen-bonding distances (d). The IR bands from (i) to (iv) can be attributable to (a)–(d), respectively. Obviously, each band is shifted to lower wavenumbers in **2** compared with **1**, indicating that the hydrogen-bonding network is especially strengthened through this powerful Ce-mediated molecular tailoring process.

Figure 4.10 In situ IR spectra of **2** under a water vapor pressure of 0.50 kPa, 1.50 kPa, and 2.50 kPa in the OH stretching region. The observed bands were well reproduced by the sum (solid lines) of four Gaussian peaks (broken lines).

Figure 4.11 The $\nu(\text{OH})$ region of water molecules obtained from in situ IR spectra of **1** and **2** under a water vapor pressure of 2.50 kPa ($P/P_0 = 0.90$). The observed bands were reproduced by the sum of four Gaussian peaks.

This part had been added into the main text as follows: “.....From this perspective, the states of water molecules in both compounds were explored through in situ IR characterization under water vapor condition, see *ACS Appl. Mater. Interfaces* **13**, 19138–19147(2021), *J. Phys. Chem. C* **111**, 8218–8227(2007). The absorption bands around 3000–3700 as well as 1600 cm^{-1} can be assigned to water molecules with OH stretching ($\nu(\text{OH})$) and HOH bending ($\delta(\text{HOH})$), respectively (Figure 4.9), see *ACS Appl. Mater. Interfaces* **13**, 19138–19147(2021), *J. Phys. Chem. C* **111**, 8218–8227(2007), *Science* **304**, 1134–1137(2004). It has been demonstrated that the band with regard to $\delta(\text{HOH})$ of water molecules remains less responsive to hydrogen-bonding networks, see *Science* **344**, 1009–1012(2014). More importantly, it is well known that the $\nu(\text{OH})$ bands of water molecules in a hydrogen bonding range occur individually and the positions of the bands reduce as the increase in the number and strength of hydrogen bonds, see *ACS Appl. Mater. Interfaces* **13**, 19138–19147(2021), *J. Phys. Chem. C* **111**, 8218–8227(2007), *J. Phys. Chem.* **93**, 4837–4843(1989). As expected, the characteristic broad bands appeared gradually after introducing water vapor (Figure 4.10). Figure 4.11 highlights the hydroxyl stretching region of the spectra, the presented bands had been deconvoluted into four Gaussian peaks

located around (i) 3513–3569 cm^{-1} , (ii) 3327–3403 cm^{-1} , (iii) 3155–3236 cm^{-1} , and (iv) 2988–3012 cm^{-1} , which may be classified into the following four groups based on the water molecules in crystal structures of **1-2** as well as previous research about the state of water molecules in Keggin-type and Preyssler-type POM systems reported by Uchida *et al.*: the $\nu(\text{OH})$ bands of water molecules (a) in the vicinity of a metal ion (Na^+ or Ce^{3+}) and without hydrogen-bonds, (b) in the vicinity of a metal ion (Na^+ or Ce^{3+}) and at a hydrogen-bonding range with an oxide ion or an oxygen atom, (c) in hydrogen-bonding distances with two oxygen atoms and/or oxide ion, and in (c) featuring short hydrogen-bonding distances (d). The IR bands from (i) to (iv) can be attributable to (a)-(d), respectively. Obviously, each band is shifted to lower wavenumbers in **2** compared with **1**, indicating that the hydrogen-bonding network has been particularly strengthened through this powerful Ce-mediated molecular tailoring process.”

Question 12: The effect of lanthanide ions on proton conduction has been explored with POM-based compounds. Please refer to *ACS Applied Mater Interfaces* 2021, 13, 19138.

Answer: Thank you for your helpful suggestions. In this paper, Uchida *et al.* concludes that 4f lanthanide ions possess high coordination number and flexible coordination environment, and these properties can be utilized to attract water molecules acted as aqua ligands and facilitate proton conduction, as well as to build robust frameworks. Inspired by Uchida and the other proton conduction based on lanthanide-containing POM-based compounds, see *J. Am. Chem. Soc.* **144**, 19603–19610(2022), *Angew. Chem. Int. Ed.* **60**, 16953–16957(2021), *Angew. Chem. Int. Ed.* **57**, 8416–8420(2018), the effect of lanthanide ions on proton conduction has been discussed. We found that $\{\text{Ce}(\text{H}_2\text{O})_7\}$ moieties with Ce ‘Outer’ binding modes and $\{\text{Ce}(\text{H}_2\text{O})_3\}$ moieties with Ce ‘On’ binding modes play great roles for building 3D framework through Ce-O \cdots O-W or Ce-O \cdots O-Ce hydrogen-bonding interactions, while $\{\text{Ce}(\text{H}_2\text{O})_x\}$ ($x = 3$ to 5) moieties with Ce ‘inner’ binding modes could stabilize the half-closed motif and its abundant aqua ligands may also participate in building hydrogen-bonding network inside the cavity, both of which are favorable for proton conduction. The related discussions and references had been added into the revised

paper as follows: "...each $\{\text{Ce}_{11}\text{Mo}_{96}\}$ in **2** is connected to seven adjacent $\{\text{Ce}_{11}\text{Mo}_{96}\}$ by twelve intermolecular atomic $\text{O}\cdots\text{O}$ distances from 2.8 to 3.1 Å, it is worth noting that these hydrogen-bonding interactions could not be formed without the participation of coordinated water on lanthanide Ce ions with Ce 'Outer' and 'On' binding modes. In fact, 4*f* ions can be utilized to attract water molecules as aqua ligands and to build robust frameworks that can facilitate proton conduction due to its high coordination number and flexible coordination environment." "...Inside $\{\text{Ce}(\text{H}_2\text{O})_x\}$ ($x = 3$ to 5) Moieties. The utilization of 9-coordinate Ce^{3+} could stabilize the half-closed motif and its abundant aqua ligands may also participate in building hydrogen-bonding network inside the cavity....".

Thanks a lot for the referee's kindly comments!

REVIEWERS' COMMENTS

Reviewer #1 (Remarks to the Author):

I have carefully read the responses by authors and their revised manuscript. I believe that they did a great job in carrying out the additional experiments (deuterated proton conductivity, stability, etc.) and supporting data. They also properly revised the manuscript and adequately addressed the concerns/comments I had in the previous round of review. Therefore, I have no additional changes to suggest and I would be glad to support its publication in the current form.

Reviewer #2 (Remarks to the Author):

The authors have adequately addressed the concerns raised in the initial submission. An independent analysis part about the synthesis discussion from {Mo₁₃₂} to {Ce₁₁Mo₉₆} had been well-presented, which had provided valuable information for further precise synthesis of polyoxometalates. The crystallinity and purity of the samples, noisy XPS spectra as well as issues of pictures or references also had been properly addressed. Furthermore, appropriate theoretical and spectroscopic evidences related to the optical characteristics and proton conduction property had been fully discussed, this part would strengthen the paper. The current work constitutes a major step towards the development of the assembly of POM clusters for the precise tailoring of the nanocluster structure to control its properties, these findings would potentially interest the readers. Therefore, this pretty good article is recommended for acceptance into Nature Communications.

Reviewer #3 (Remarks to the Author):

Authors have revised this manuscript sufficiently. I think it has reached the level of publication in this journal. I recommend accepting the paper.

Reviewer #4 (Remarks to the Author):

Thank you for revising the manuscript. The comments from the reviewers have been appropriately addressed. I have one additional comment. Since the compounds contain redox-active Mo atoms, the authors should evaluate the possible contribution of electron conduction by measuring DC conductivity.

Reviewer #1 (Remarks to the Author)

I have carefully read the responses by authors and their revised manuscript. I believe that they did a great job in carrying out the additional experiments (deuterated proton conductivity, stability, etc.) and supporting data. They also properly revised the manuscript and adequately addressed the concerns/comments I had in the previous round of review. Therefore, I have no additional changes to suggest and I would be glad to support its publication in the current form.

Thanks a lot for the referee's kindly comments!

Reviewer #2 (Remarks to the Author)

The authors have adequately addressed the concerns raised in the initial submission. An independent analysis part about the synthesis discussion from $\{\text{Mo}_{132}\}$ to $\{\text{Ce}_{11}\text{Mo}_{96}\}$ had been well-presented, which had provided valuable information for further precise synthesis of polyoxometalates. The crystallinity and purity of the samples, noisy XPS spectra as well as issues of pictures or references also had been properly addressed. Furthermore, appropriate theoretical and spectroscopic evidences related to the optical characteristics and proton conduction property had been fully discussed, this part would strengthen the paper. The current work constitutes a major step towards the development of the assembly of POM clusters for the precise tailoring of the nanocluster structure to control its properties, these findings would potentially interest the readers. Therefore, this pretty good article is recommended for acceptance into Nature Communications.

Thanks a lot for the referee's kindly comments!

Reviewer #3 (Remarks to the Author)

Authors have revised this manuscript sufficiently. I think it has reached the level of publication in this journal. I recommend accepting the paper.

Thanks a lot for the referee's kindly comments!

Reviewer #4 (Remarks to the Author)

Thank you for revising the manuscript. The comments from the reviewers have been appropriately addressed.

Question: I have one additional comment. Since the compounds contain redox-active Mo atoms, the authors should evaluate the possible contribution of electron conduction by measuring DC conductivity.

Answer: Thank you for your helpful suggestions. We investigated the electron conductivity of both compounds by direct-current (DC) measurements, the current is as low as 0.2 nA at 0.1 V, thus the electron conductivity was negligible ($R > 10^8 \Omega$), see *Chem. Sci.* 10, 556–563(2019). The DC conductivity was conducted on a 4-Point Probes Resistivity Measurement System (RTS-9). In fact, the electron conductivity (direct-current experiment) measurement had already been mentioned during the last revision in the main text as follows: “The electron conductivity (direct-current experiment) and ionic conductivity (deuterated water experiment) measurements confirmed that the conduction is based on the transport of protons”.

Thanks a lot for the referee’s kindly comments!